# Comparison of Chemical Compositions, Antioxidant, and Anti-Photoaging Activities of *Paeonia suffruticosa* Flowers at Different Flowering Stages

**DOI:** 10.3390/antiox8090345

**Published:** 2019-09-01

**Authors:** Jingyu He, Yaqian Dong, Xiaoyan Liu, Yiling Wan, Tanwei Gu, Xuefeng Zhou, Menghua Liu

**Affiliations:** 1Bioengineering Research Centre, Guangzhou Institute of Advanced Technology, Chinese Academy of Sciences, Guangzhou 511458, China; 2Guangdong Provincial Key Laboratory of New Drug Screening, School of Pharmaceutical Sciences, Southern Medical University, Guangzhou 510515, China; 3CAS Key Laboratory of Tropical Marine Bio-Resources and Ecology, South China Sea Institute of Oceanology, Chinese Academy of Sciences, Guangzhou 510301, China; 4Guangdong Key Laboratory of Marine Materia Medica, South China Sea Institute of Oceanology, Chinese Academy of Sciences, Guangzhou 510301, China

**Keywords:** *Paeonia suffruticosa*, flower, antioxidant activity, anti-photoaging activity, phytochemicals, chemometric analysis

## Abstract

*Paeonia suffruticosa* is an ornamental, edible, and medicinal plant. The ethanolic extracts of *P*. *suffruticosa* bud and flower were examined for their antioxidant, anti-photoaging, and phytochemical properties prior to chemometric analysis. The results showed that the bud ethanolic extract (BEE) and the flower (the early flowering stage) ethanolic extract (FEE) had better antioxidant activities, and significantly increased the activities of superoxide dismutase (SOD) and glutathione peroxidase (GSH-Px) and reduced the levels of tumor necrosis factor-α (TNF-α) and interleukin-6 (IL-6) in the skin tissues. In total, 68 compounds, including 20 flavonoids, 15 phenolic derivatives, 12 terpenoids, 9 fatty acids, and 12 others were identified or tentatively identified by ultra-fast liquid chromatography quadrupole time-of-flight mass spectrometry (UFLC-Q-TOF-MS). Gallic acid, 1,2,3,4,6-*O*-pentagalloyl glucose, paeoniflorin, and oxypaeoniflorin were predominant compounds in the extracts. Taken together, *P*. *suffruticosa* flowers are a candidate for functional material in food and health related industries, and their optimal time to harvest is before the early flowering stage.

## 1. Introduction

Ultraviolet-B (UVB) radiation is one of the most effective constituents of solar light, and has become the primary source of oxidative stress to humans. UVB generally impacts the basal layer of epidermal skin and causes premature skin photoaging, local and systemic immunosuppression, cutaneous inflammatory disorders, and photocarcinogenesis [1]. It is concluded that oxidative stress is a problem of skin cells and that endogenous and exogenous antioxidants could play an important role in decreasing it [2]. A large number of phytochemicals obtained from plants could reduce the harmful effects of oxidative stress and help to prevent the photoaging of skin [3,4]. Experimental and epidemiological studies indicated that the consumption of plant foods is related to lower incidence of age-related diseases [5]. Thus, there is increasing interest in the antioxidant properties of phytochemicals found in plant foods.

*Paeonia suffruticosa*, belonging to Paeoniaceae, is mainly distributed in China, Japan, America, and Europe, and is an ornamental, edible, and medicinal plant in China and Japan [6,7]. Root bark of *P*. *suffruticosa*, named Cortex Moutan, is an important Chinese traditional medicine with the functions of lowering blood sugar, lowering blood pressure, anti-inflammatory, anti-bacterial, anti-tumor, and regulating the cardiovascular system [8]. *P. suffruticosa* flower as a characteristic natural resource that also has a long history of medicinal and edible use in China. Monoterpenoids, flavonoids, and essential oil have been found in *P. suffruticosa* flower [6,9,10]. The essential oils of *P. suffruticosa* flower buds possesses an inhibitory effect on common food-borne bacterial pathogens [6]. Moreover, *P. suffruticosa* flower as medicinal herb is used for the treatment of gynaecological diseases, and has experienced a growing number of applications in the food industry to produce cake, herbal tea, and drink, as well as in the cosmetic industry to produce facial masks and sunscreen creams. It is known that the chemical constituents that are responsible for the bioactivity are affected by the different flowering stages [11,12]. However, the characteristic phytochemicals and the composition change in the flowers are unclear and there is no principle to guide as to which flowering stage is suitable for harvesting when the flower is used in industry. The study on the relationship between phytochemicals and bioactivity during the flowering stages is less documented, which is a limitation to the development of products originated from the *P. suffruticosa* flower.

In this study, the antioxidant activities of the ethanolic extracts of *P. suffruticosa* flowers were evaluated using four different methods, including 2,2-diphenyl-1-picrylhydrazyl (DPPH) and hydroxyl radical scavenging activity, inhibition of β-carotene bleaching, and ferric reducing antioxidant power (FRAP) assays. The anti-photoaging activity of the ethanolic extracts of *P. suffruticosa* flowers was evaluated using a UVB-irradiated mouse model. The characteristic chemicals of the ethanolic extract of *P. suffruticosa* flowers were investigated prior to the correlation analysis between the multiple ingredients and their bioactivities.

## 2. Materials and Methods

### 2.1. Plant Materials

*P. suffruticosa* flowering buds were collected on 30 March 2016, and *P. suffruticosa* flowers at the early flowering stage were collected ten days later. *P. suffruticosa* flowers at the full flowering stage was collected on 20 April 2016, and then divided into petal and stamen (Figure 1). All of *P. suffruticosa* samples were collected in Luoyang, Henan province, China. All voucher specimens were stored at the Guangzhou Institute of Advanced Technology, Chinese Academy of Sciences, Guangdong, China. 

### 2.2. Chemicals and Reagents

Leucine, gallic acid, adenosine, tryptophan, and *p*-hydroxybenzoic acid were purchased from the National Institute for the Control of Pharmaceutical and Biological Products (Beijing, China). Morroniside, loganin, and geniposide were purchased from Chengdu Biopurify Phytochemicals Ltd. (Chengdu, China). Paeoniflorin, oxypaeoniflorin, benzoylpaeoniflorin, paeonol, 1,2,3,4,6-*O*-pentagalloyl glucose, luteolin, and apigenin were purchased from Chengdu Pufei De Biotech Co., Ltd. (Chengdu, China). Quercetin-3-*O*-glucoside and rhoifolin were purchased from Sigma-Aldrich (Shanghai, China). Hexadecanoic acid, 9-octadecenoic acid, octadecanoic acid, and 2,4,6-tris(2-pyridyl)-1,3,5-triazine were purchased from Shanghai Yuan Ye Biotechnology Co., Ltd. (Shanghai, China). 2,2-diphenyl-1-picrylhydrazyl (DPPH), butylated hydroxytoluene (BHT), β-carotene, and vitamin C (VC) were purchased from Shanghai Ekear Bio-Tech Co., Ltd. (Shanghai, China). HPLC-grade acetonitrile and LC/MS grade methanol were purchased from Fisher Scientific (Fair Lawn, NJ, USA). Water was obtained from an ultrapure water system (Purelab Plus, Pall, Port Washington, NY, USA). 

### 2.3. Sample Preparation

The dried and ground sample (100 g) was extracted with 95% (*v/v*) ethanol (800 mL) in a KQ600DE ultrasonic bath (Kunshan, Jiangsu, China) for 30 min at room temperature. This extraction was repeated after filtration. The combined filtrate was evaporated under vacuum to yield the ethanolic extract. The dried ethanolic extract was weighed accurately and dissolved in 95% ethanol to obtain a series of solutions with different concentrations for the further study.

### 2.4. Total Phenolic Content Assay

Total phenolic content was determined using the described method [13]. Sample solution (1.0 mL) was mixed with the Folin–Ciocalteu reagent (5.0 mL; 0.1 mmol/L) and allowed to incubate for 5 min. A sodium carbonate solution (4.0 mL; 75 g/L) was added, and the mixture was vortexed and incubated at 25 °C for 30 min in darkness. A standard gallic acid solution was used and the absorbance was determined at 765 nm by spectrophotometer (UV-6100; Shanghai Metash Instrument Co., Ltd., Shanghai, China). Total phenolic content was expressed as mg gallic acid equivalents per gram of extract (mg GAE/g ext.). All measurements were performed in triplicate.

### 2.5. Total Flavonoid Content Assay

Total flavonoid content was measured using a previously reported method [13]. In this procedure, a mixture of an aliquot (2.0 mL) of sample solution, a 95% ethanol solution (4.0 mL), and sodium nitrite solution (2.0 mL; 50 g/L) was incubated at 25 °C for 5 min, and aluminium nitrate solution (2.0 mL; 100 g/L) was then added. After incubation at 25 °C for 6 min, sodium hydroxide solution (5.0 mL; 4 g/L) was added. Rutin was used to calculate the standard curve, and the absorbance was determined at 510 nm by spectrophotometer. Total flavonoid content was expressed as mg rutin equivalent per gram of extract (mg RE/g ext.). All measurements were performed in triplicate.

### 2.6. Antioxidant Assay

#### 2.6.1. DPPH Radical Scavenging Activity Assay

DPPH radical scavenging activity was measured according to a previously reported method [14]. Briefly, sample solution (1.0 mL) was mixed with a freshly prepared methanol solution of DPPH (1.0 mL; 80 μg/mL). The absorbance was measured at 517 nm spectrophotometrically after the mixture was incubated for 30 min at 25 °C. VC and BHT were used as two standard antioxidant compounds. All determinations were performed in triplicate. In control experiment, 95% ethanol (2.0 mL) was used. The radical scavenging capability of DPPH was calculated using the following equation: Scavenging effect (%) = (1 − *A_sample_*/*A_control_*) × 100, where *A_sample_* and *A_control_* are the absorbance of the sample and the control, respectively.

#### 2.6.2. Hydroxyl Radical Scavenging Activity Assay

The hydroxyl radical scavenging activity was determined according to a previously described method [13]. The hydroxyl radical was produced by the Fenton reaction between ferrous sulphate and hydrogen peroxide. Briefly, a sample solution (2.0 mL), a ferrous sulphate solution (2.0 mL; 5.0 mmol/L), a salicylic acid solution (2.0 mL; 5.0 mmol/L), and a hydrogen peroxide solution (2.0 mL; 5.0 mmol/L) were added to the test tubes. The absorbance was measured spectrophotometrically at 510 nm after the mixture was incubated at 37 °C for 1 h. VC was used as the standard antioxidant compound. In control experiment, 95% ethanol (2.0 mL) was used. All determinations were performed in triplicate. The hydroxyl radical scavenging capability was calculated using the following equation: Scavenging effect (%) = [1 − (*A_sample_* − *A_-hydrogen peroxide_*)/*A_control_*] × 100, where *A_sample_*, *A_-hydrogen peroxide_*, and *A_control_* are the absorbance of the sample, the sample without hydrogen peroxide, and the control, respectively.

#### 2.6.3. Ferric Reducing Antioxidant Power (FRAP) Assay

In this study, FRAP assays were performed according to a previously described method [14]. The antioxidants could restore ferric tripyridyltriazine (Fe^3+^-TPTZ) to blue Fe^2+^-TPTZ under acidic conditions. The FRAP solution was prepared by mixing acetate buffer (300 mM, pH 3.6), TPTZ (10 mM) and FeCl_3_ solution (20 mM) in a ratio of 10:1:1 (*v/v/v*). An aliquot (0.15 mL) of sample solution was mixed with the FRAP solution (2.85 mL) and incubated at 37 °C for 30 min. The total antioxidant activity of the samples could be evaluated by the absorbance of Fe^2+^-TPTZ at 593 nm. All measurements were performed in triplicate. The results of the FRAP assay were reported in mM FeSO_4_.

#### 2.6.4. Inhibition of β-Carotene Bleaching Assay

Inhibitory ability of β-carotene bleaching was assessed according to a previously described method [14]. Briefly, a solution of β-carotene was prepared by dissolving 2 mg in chloroform (10 mL). Two millilitres of this solution was pipetted into a round-bottom flask. After the chloroform was removed at 40 °C under vacuum, linoleic acid (40 mg), Tween-80 emulsifier (400 mg), and distilled water (100 mL) were added to the flask with vigorous shaking. An aliquot (4.8 mL) of this emulsion was transferred to different test tubes containing the sample solution (0.2 mL). A 4.8 mL portion of the emulsion combined with 95% ethanol (0.2 mL) was used as a control. The tubes were shaken and incubated at 50 °C in a water bath. As soon as the emulsion was added to each tube, the absorbance was measured at 470 nm at 0 min and at 15-min intervals over a 120 min period. All measurements were performed in triplicate. The β-carotene bleaching inhibition was calculated using the following equation: Bleaching inhibition (%) = [1 − (*A_sample–0_* − *A_sample–t_*)/(*A_control–0_* − *A_control–t_*)] × 100, where *A_sample–0_* and *A_control–0_* are the initial absorbance (t = 0 min) of the sample and the control, respectively; *A_sample–t_* and *A_control–t_* are the absorbance (t = 15, 30, 45, 60, 75, 90, 105, and 120 min) of the sample and the control, respectively.

### 2.7. Animal Treatment and UV Irradiation

Kunming mice (body weights of 18–20 g) were purchased from the Experimental Animal Center of Southern Medical University (Guangzhou, China). Mice were fed on standard laboratory diet and water at libitum, and kept in 12 h dark/light cycle room at 21 ± 3 °C with a relative humidity of 55% ± 10% for one week prior to the ultraviolet treatment. Animal experiments and all procedures were approved by the Animal Care and Use Committee of Southern Medical University (Guangzhou, China). Mice were randomly divided into 7 groups (n = 6), control group (no irradiation), model group (irradiated group with no treatment), model + emulsifier (ME) group (irradiated group treated with emulsifier containing stearic acid 10.0 g, glycerin monostearate 3.0 g, liquid paraffin 5.0 g, vaseline 1.0 g, lanolin 4.0 g, triethanolamine 1.0 g, and distilled water 50 mL), bud ethanolic extract (BEE) group (irradiated group treated with 2.0% (*w/w*) BEE in emulsifier), flower ethanolic extract (FEE) group (irradiated group treated with 2.0% (*w/w*) FEE in emulsifier), petal ethanolic extract (PEE) group (irradiated group treated with 2.0% (*w/w*) PEE in emulsifier), stamen ethanolic extract (SEE) group (irradiated group treated with 2.0% (*w/w*) SEE in emulsifier). The shaved dorsal skin of the mice was exposed to 120 mJ/cm UVB radiations (emission peak 306 nm; Sankyo Denki Co., Tokyo, Japan) for 20 min, thrice a week. The preparations (0.2 g) were topically administered daily to the back of the mouse skin. Mice were anaesthetized and their dorsal skin tissues were collected at the end of experiment (4 weeks). Dorsal skin tissues were washed with physiological saline (0.9% NaCl) and stored at −80 °C until analyses were performed. Half of dorsal skin tissue was used for histological analysis, and the other half was used for biochemical analysis.

### 2.8. Histological Analysis

Dorsal mouse skin samples were fixed overnight with 4% paraformaldehyde in phosphate buffered saline (PBS) at 4 °C and then embedded in paraffin. Paraffin sections (4 μm) were mounted on silane-coated slides and stained with Hematoxylin&Eosin (H&E) using H&E staining kit (Sangon Biotech (Shanghai) Co., Ltd., Shanghai, China). The sections were examined, and images were recorded using a microscope and Image-Pro Express 5.1.1.14 Pathology Image Analysis System (Olympus Corporation, Tokyo, Japan) at 400× magnifications. The epidermal thickness was measured at 8 random sites randomly selected locations per slide using the image analysis program.

### 2.9. Biochemical Analysis

The dorsal skin tissues were homogenized in a glass homogenizer with a buffer containing 1.15% KCl in a 1:10 (*w/v*) whole homogenate. The homogenates used for the biochemical analysis were obtained by centrifugation at 12,000× *g* for 30 min. Protein levels in skin homogenates were measured using the Pierce BCA protein assay kit (Beyotime Biotechnology, Shanghai, China). Concentrations of superoxide dismutase (SOD), glutathione peroxidase (GSH-Px), tumor necrosis factor-α (TNF-α), and interleukin-6 (IL-6) in dorsal skin tissues were determined using mouse-specific enzyme-linked immunosorbent assay (ELISA) kits (NeoBioscience, Shenzhen, China). Analysis was performed according to the manufacturer’s instruction.

### 2.10. UFLC-DAD-Q-TOF-MS Analysis

#### 2.10.1. System and Conditions

Ultra-fast liquid chromatography (UFLC) analysis was performed on a Shimadzu UFLC XR system (Shimadzu Corp., Kyoto, Japan) with an Agilent Eclipse Plus C_18_ column (2.1 i.d. ×100 mm, 1.8 μm, Agilent Technologies, CA, USA). The mobile phases were composed of methanol (A) and water with 0.1% formic acid (B) using a linear gradient elution of 5–100% A within 30 min at 0.3 mL/min. The injection volume was 5 μL, and the column temperature was set at 25 °C. The identification experiment was performed using AB SCIEX Triple TOF 5600 plus mass spectrometer system (AB SCIEX, Foster City, CA, USA). The system was operated using the Analyst TF 1.6 software (AB SCIEX, Foster City, CA, USA). The parameters of the MS detector were as follows: Ion source gas 155 psi; ion source gas 255 psi; curtain gas 30 psi; source temperature 550 °C; ion spray voltage floating 4500 V; collision energy 35 eV; collision energy spread 15 eV; and declustering potential 80 eV. Spectra were acquired in a scan range from *m/z* 100–1500. Both positive and negative ion modes were used for compounds ionization. Nitrogen was used as the nebulizer and auxiliary gas.

#### 2.10.2. Establishment of Tentative Peak Assignment

The UFLC-Q-TOF-MS data of samples were extracted and analysed using the PeakView software (AB SCIEX, Foster City, CA, USA), mainly with the XIC manager tool, which provided the quasi-molecular weights, mass errors, and isotope pattern fits. The predicted formula with errors less than ± 5 ppm was searched against the compounds reported in the genus *Paeonia* to obtain the tentative identification. The identification of the compounds within the samples was further confirmed by determining the possible elemental compositions of the fragment ions and the proposed fragmentation pathways using their MS spectrum.

### 2.11. HPLC-DAD Analysis

High performance liquid chromatography (HPLC) analysis was performed on an Agilent (CA, USA) 1290 UPLC-DAD system with a reverse-phase Agilent EC-C_18_ column (3.0 i.d. × 100 mm, 2.7 μm, Agilent Technologies, CA, USA). A gradient elution mobile phase system composed of 0.5% formic acid solution (A) and acetonitrile (B) was applied as follows: 0–3 min, 5–8% B; 3–4 min, 8–13% B; 4–13 min, 13–14% B; 13–14 min, 14–20% B; 14–29 min, 20–30% B; 29–30 min, 30–50% B; 30–33min, 50–95%B. The mobile phase flow rate was 0.3 mL/min, and the column temperature was maintained at 30 °C. The injection volume was 5 μL and the UV wavelength was set at 254 nm.

The developed method was validated by linear range, limit of detection (LOD), limit of quantitation (LOQ), precision, and recovery according to the previous report [14]. 

### 2.12. Statistical Analysis

The results were expressed as the mean ± standard deviation (SD) of each experiment. The data obtained were analysed using a one-way analysis of variance (ANOVA) followed by Tukey’s multiple comparison test. A *p* value < 0.05 was considered to be significant. Pearson’s correlation analysis was used to characterize the correlation between the phytochemicals and the bioactivities. One-way ANOVA and Pearson’s correlation analysis were performed using the software IBM SPSS Statistics (version 19.0).

## 3. Results and Discussion

### 3.1. Total Phenolic and Flavonoid Contents

The extract yields of bud, flower, petal, and stamen of *P. suffruticosa* had obvious differences (Table 1). Total phenolic contents in the BEE, FEE, PEE, and SEE were determined. BEE had the highest total phenolic content (191.80 ± 3.44 mg GAE/g ext.), followed by FEE (113.84 ± 0.84 mg GAE/g ext.), SEE (77.25 ± 0.58 mg GAE/g ext.) and PEE (51.66 ± 0.34 mg GAE/g ext.). Unexpectedly, FEE had the highest total flavonoid content (49.87 ± 0.82 mg RE/g ext.), followed by BEE (30.60 ± 0.94 mg RE/g ext.), PEE (12.98 ± 0.78 mg RE/g ext.) and SEE (8.86 ± 0.98 mg RE/g ext.). Total phenolic content in the flowers had a trend of gradual decrease during the flowering stage of *P. suffruticosa*. In the other hand, there might be a maximum content of flavonoids in the flowering stage.

### 3.2. Antioxidant Activity In Vitro

The antioxidant capabilities of BEE, FEE, PEE, and SEE were tested by the DPPH and hydroxyl radical scavenging, FRAP, and inhibition of β-carotene bleaching assays (Figure 2). The FEE had stronger DPPH radical scavenging activity than the other extracts and BHT below 50 μg/mL, while it was obviously weaker than that of VC at concentration 10–200 μg/mL. The BEE and SEE also had stronger activity than BHT. On the basic of IC_50_ values, antioxidant activity is defined as the concentration of antioxidant required for 50% scavenging of the radicals (Table 1). The IC_50_ value of FEE for scavenging DPPH was 14.83 ± 2.03 μg/mL, which was lower than that of BEE (34.44 ± 1.35 μg/mL), PEE (64.00 ± 5.60 μg/mL), and SEE (42.70 ± 0.79 μg/mL). A lower IC_50_ value suggested that antioxidant activity was better in the four extracts. Unexpectedly, the IC_50_ value of BEE was 1.57 ± 0.04 mg/mL, which was lower than that of FEE (1.79 ± 0.02 mg/mL), PEE (2.57 ± 0.06 mg/mL), and SEE (2.10 ± 0.004 mg/mL). The BEE had strongest hydroxyl radical scavenging activity than the other extracts, but was weaker than VC. In the FRAP assay, FEE had the strongest activity, followed by BEE, PEE, and SEE. Moreover, FEE had similar activity to BHT, but was much stronger than VC. The activity of BEE in the inhibition of the β-carotene bleaching assay was a little weaker than that of BHT (100 μg/mL) over 120 min, but had stronger and longer lasting inhibitory activity than that of other extracts at 100 μg/mL. Based on the results of the antioxidant evaluation, the antioxidant activities of BEE and FEE were noticeable. BEE had a greater effect on hydroxyl radical scavenging and inhibition of the β-carotene bleaching, whereas FEE had a greater effect on DPPH radical scavenging and FRAP. These results indicated that there was a correlation between bioactivity and the flowers at the different harvest time, due to the change of chemical compositions. 

### 3.3. Effects on the Morphology in Mouse Skin 

Skin that is exposed excessively to UVB irradiation is easily damaged. UVB irradiation led to excessive damage to the morphology and integrity of the mouse skin. It is reported that UVB radiation on the skin induces a variety of responses in the epidermis, including keratinocyte proliferation that leads to epidermal hyperplasia and thickening [15]. In this study, the change incurred on the mouse skin was assessed after UVB irradiation. The epidermal thickness of the ME group significantly increased by 1.14 fold when compared to that of the control group. In the case of the extract treatment groups, the epidermal thickness of BEE, FEE, PEE, and SEE groups significantly decreased (*p* < 0.001) to 70.62%, 74.62%, 88.13%, and 90.74%, respectively, when compared to that of the ME group (Figure 3H). Topical treatment with the four extracts had the effect of inhibiting the epidermal thickening, and of which BEE could best prevent UVB-induced damage to the skin. A lot of studies have reported natural products are the important resources of antioxidants [16,17]. Phenols and flavonoids, the main constituents in flowers, were considered as antioxidants and UV absorbing secondary metabolites [18,19]. It has also been reported that flavonoids help to prevent the skin photoaging by regulating pro-inflammatory cytokines [20,21]. Interestingly, abundant phenols and flavonoids were found in the BEE and FEE, respectively. Phenols and flavonoids in the BEE and FEE appear to be useful against the chronic effects of UV light. More studies on antioxidant and anti-inflammatory activities of the four extracts were performed in the next experiments. 

### 3.4. Effects on the Activities of SOD and GSH-Px in Mouse Skin 

The activities of endogenous antioxidants in the skin tissues were further investigated (Figure 4A,B). The activity of SOD in the model group and model + emulsifier groups was significantly decreased when compared with the control group. Using a UVB-irradiated mouse model, topical treatment with BEE and FEE could significantly increase (*p* < 0.05) the activity of SOD, but topical treatment with PEE and SEE had no significant effect on the activity of SOD. The activity of GSH-Px in the skin tissues was significantly decreased when the mice were treated with UVB irradiation. Topical treatment with BEE (*p* < 0.001), FEE (*p* < 0.001), PEE (*p* < 0.001), and SEE (*p* < 0.05) significantly increased the activity of GSH-Px in the skin of UVB irradiated mice. 

UVB irradiated skin depletes antioxidant defensive capabilities [5]. ROS plays a significant role in UVB-induced skin carcinogenesis. In the body, endogenous antioxidants offset UV-induced oxidative stress by the neutralization of the ROS prior to oxidative changes occuring in the tissues [22]. SOD could alternately catalyze the dismutation of the superoxide radical into either ordinary molecular oxygen or hydrogen peroxide, and thereby is an important antioxidant defense [23]. GSH, a free radical-scavenger and a cofactor for protective enzymes, plays a pivotal role in protecting cells from oxidative damage [24]. Normally, these enzymes are able to scavenge the ROS efficiently, and consequently protect skin from damage. However, excessive and chronic exposure to UVB radiation can overwhelm the cutaneous antioxidant capacity leading to oxidative damage, resulting in skin photoaging [25]. BEE and FEE with excellent antioxidant capability could significantly increase the activities of SOD and GSH-Px in the damaged skin, which was responsible for inhibiting the epidermal thickening, consequently ameliorating the skin’s pathological symptom. 

### 3.5. Effects on the Leveles of TNF-α and IL-6 in Mouse Skin 

As Figure 4C,D shows, inflammatory cytokines (TNF-α and IL-6) in the UVB irradiated group were significantly increased when compared with the control group. Topical treatment with BEE (*p* < 0.001), FEE (*p* < 0.001), PEE (*p* < 0.001), and SEE (*p* < 0.01) significantly decreased the content of TNF-α in the skin tissues of UVB irradiated mice. BEE and FEE could significantly decrease the content of IL-6 in the skin tissues of UVB irradiated mice. Contrarily, no significant effect on the contents of IL-6 was found in PEE and SEE treated groups. 

Inflammation response that causes erythema, edema, and an influx of inflammatory cells, is one of the most obvious results of UVB radiation. Epidermal keratinocytes strongly contribute to cutaneous inflammation via their release of pro-inflammatory cytokines [26]. Pro-inflammatory cytokines, such as TNF-α and IL-6, play crucial roles in inflammatory development and are considered to be indicators of the degree of inflammation [27]. Moreover, pro-inflammatory cytokines stimulate the epidermal keratinocytes and dermal fibroblasts, and degrade dermal collagen and elastic fibers, consequently causing the epidermal thickening and the formation of wrinkles [4]. Accordingly, the inhibition of pro-inflammatory cytokines secretion induced by UVB irradiation is important to protect skin from photo-damage. BEE and FEE could significantly decrease the levels of TNF-α and IL-6 in damaged skin and thereby partly compensate for the inflammation induced by UVB radiation. As a result, the anti-inflammatory capability of BEE and FEE was responsible for ameliorating the skin’s pathological symptom. These results suggested that the BEE and FEE, with more excellent antioxidant and anti-inflammatory capabilities, had a greater effect on the prevention of skin photoaging. Their molecular mechanism of anti-photoaging is still unclear and is, thus, required in future study.

### 3.6. Chemical Compositions of Ethanolic Extracts of P. suffruticosa Flowers

Compounds at the different flowering stages of *P. suffruticosa* were identified by UPLC-Q-TOF-MS/MS method through comparison retention time (Rt), accurate molecular weight (MW), and fragmentography with the reference standards. When no standards were unavailable, compounds were tentatively identified by the accurate molecular weight and fragmentation pathway, which had been carried out on the same kind of compounds or literature report. As shown in Table 2, a total of 68 compounds, including 12 monoterpenoids, 20 flavonoids, 15 phenols and their derivatives, 9 fatty acids, and 12 others, were identified or tentatively identified in this study. 

Monoterpenoids are present as ubiquitous chemical compounds in *P. suffruticosa*, which are characterized by two isoprene units and have the molecular formula C_10_H_16_. Among them, paeoniflorin (**28)**, oxypaeoniflorin (**21**), and benzoylpaeoniflorin (**54**) were precisely identified by the comparison to their reference standards, respectively. Analyzing the mass spectrogram, the fragment ions were usually generated by losing the units of glucose residue (C_6_H_10_O_5_, 162 Da), benzoic acid residue (C_7_H_6_O_2_, 122 Da), and hydroxybenzoic acid residue (C_7_H_6_O_3_, 138 Da). Therefore, compound **3** with *m/z* 375.12986 (calculated to be C_16_H_24_O_10_), compound **17** with *m/z* 527.14061 (calculated to be C_23_H_28_O_14_), compound **23** with *m/z* 525.16213 (calculated to be C_24_H_29_O_13_), compound **30** with *m/z* 647.16060 (calculated to be C_30_H_32_O_16_), compound **36** with *m/z* 631.16737 (calculated to be C_30_H_32_O_15_), compound **42** with *m/z* 615.17285 (calculated to be C_30_H_32_O_14_), compound **49** with *m/z* 599.17976 (calculated to be C_30_H_32_O_13_), compound **51** with *m/z* 599.18003 (calculated to be C_30_H_32_O_13_), and compound **52** with *m/z* 629.19181 (calculated to be C_31_H_34_O_14_) in negative mode were tentatively identified as 8-debenzoylpaeoniflorin [28], debenzoygalloypeaoniflorin, mudanpioside E, galloyoxypaeoniflorin, galloylpaeonifflorin, benzoyloxypaeoniflorin, mudanploside H, mudanpioside C, benzoloxypaeoniflorin, and mudanpioside J, respectively [29].

Flavonoids, especially quercetin, kaempferol, isorhamnetin, apigenin and their derivatives, are another set of components also identified in *P. suffruticosa* [30]. Their fragment behaviors have been investigated very well in positive and negative ion modes. Usually, the [aglycone−H]^−^ ion is generated by loss of the linkage of sugars and thereafter followed by Ret–Diels–Alder (RDA) dissociation, subsequently generating *m/z* 179 and 151 ions [31]. Compounds **46**, **53**, and **55** were undeniably identified as rhoifolin, luteolin, and apigenin, respectively. According to the fragmentation mechanism, compound **45** were inferred as apigenin-7-*O*-glucoside with *m/z* at 431.10019 in negative ion mode [30]. Kaempferol is an isomer of luteolin with molecular formula C_15_H_10_O_6_. Compounds **29**, **39**, **41**, and **43** were inferred as kaempferol-3,7-*O*-di-glucoside, kaempferol-3-*O*-glucoside, kaempferol-3-*O*-rutinoside, and kaempferol-3-*O*-(2′′-*O*-galloyl)-glucoside [10,30]. Isorhamnetin with molecular formula C_16_H_12_O_7_, was 30 Da (CH_2_O) higher than luteolin. Compounds **44** and **48** were isomers with molecular formula of C_22_H_22_O_12_. The fragment ion at *m/z* 317 was yielded by loss of glucose. Thus they were tentatively identified as isorhamnetin-3-*O*-glucoside and isorhamnetin-7-*O*-glucoside, respectively. Compound **34** gave an [M−H]^−^ ion at *m/z* 639.15895 (calculated as C_28_H_31_O_17_) and was inferred as isorhamnetin-3,7-di-*O*-glucoside [32]. Quercetin was C_15_H_10_O_7_, 16 Da (O) higher than luteolin. Compound **40** was quercetin-3-*O*-glucoside with an [aglycone−H]^−^ ion at *m/z* 301.0354 by loss of glucose residue [10]. Compounds **26** and **33** were isomers, which gave [M−H]^−^ at *m/z* 625 and were tentatively identified as quercetin-3,7-di-*O*-glucoside or its isomer [32]. Compound **37** was 30 Da (CH_2_O) higher than compound **34** and was inferred as 6,3′-dimethoxyquercetin-7-*O*-di-glucoside with *m/z* at 669.16952. Compound **22** (eriodictyol-7-*O*-glucoside) was calculated as C_21_H_22_O_11_ and generated a fragment ion at *m/z* 287 by loss of one glycoside. Compound **32** (patuletin-3,5-di-*O*-glucoside) was calculated as C_28_H_32_O_18_, and generated a fragment ion at *m/z* 331 by loss of two glycosides [32]. Compound **38** was inferred as monoxerutin with a quasi-molecular ion at *m/z* 653.17535. Diosmin (**47**) gave a quasi-molecular ion at *m/z* 607.17007, and subsequently generated an ion of 299 by loss of a rutinoside. Compound **56** has 30 Da (CH_2_O) more than apigenin, and it was identified as chrysoeriol with an [M−H]^−^ ion at *m/z* 299.05704.

There were 14 phenols and derivatives found in the four extracts of *P. suffruticosa* flowers. Compounds **5**, **15**, **35,** and **50** were directly identified as gallic acid, *p*-hydroxybenzoic acid, 1,2,3,4,6-*O*-pentagalloyl glucose, and paeonol by the comparison of their reference standards, respectively. Compounds **4** and **9** were a pair of isomers, which showed 162 Da more than gallic acid, suggesting the presence of C_6_H_10_O_5_. They were tentatively identified as glucogallin or isomer. Interestingly, compound **10** gave 324 Da more than gallic acid, suggesting the presence of two C_6_H_10_O_5_ [33]. Compound **16** showed CH_2_ (14 Da) more than gallic acid, and was referred to be methyl gallate [29]. In mass spectrum, compound **12** was determined to be C_14_H_18_O_9_ and yield an ion at *m/z* 167.0341 [M−H−C_6_H_10_O_5_]^−^. It was tentatively identified as mudanoside A [29]. Compound **14** gave an [M−H]^−^ ion at *m/z* 315.07227, corresponding to C_13_H_16_O_9_. It yielded daughter ions by successive losses of C_6_H_10_O_5_, CO_2_ and was inferred to be gentisic acid-5-*O*-glucoside. Compounds **24** and **31** were tentatively identified as trigalloyl glucose and teragalloyl glucose. Compound **20** was an isomer of trigalloyl glucose with a characteristic structure of C_7_H_6_O_3_. Compound **25** gave an [M+H]^+^ ion at *m/z* 155.07034, corresponding to C_8_H_10_O_3_, and was tentatively identified as 3,4-dimethoxyphenol. Compound **27** was deduced as paeonolide based on the deprotonated ion at *m/z* 299.1206 and a fragment ion at *m/z* 167.0707. 

Long chain fatty acid often provides ions by losses of CO_2_, H_2_O, and (CH_2_)_n_ in MS spectrum. Compounds **64**, **66,** and **68** were accurately identified as hexadecanoic acid, 9-octadecenoic acid, and octadecanoic acid by comparing them with their reference standards. Compound **57** with *m/z* 327.21889 (calculated to be C_18_H_32_O_5_), compound **58** with *m/z* 329.23479 (calculated to be C_18_H_34_O_5_), compound **59** with *m/z* 287.22468 (calculated to be C_16_H_32_O_4_), compound **60** with *m/z* 293.21325 (calculated to be C_18_H_30_O_3_), compound **61** with *m/z* 295.22859 (calculated to be C_18_H_32_O_3_), and compound **65** with *m/z* 355.32323 (calculated to be C_22_H_44_O_3_) were tentatively identified as malyngic acid, 9,12,13-trihydroxyoctadec-10-enoic acid, 3,12-dihydroxyhexadecanoic acid, 9-oxooctadeca-10,12-dienoic acid, 13-hydroxy-9,11-octadecadienoic acid, and 2-hydroxybehenic acid, respectively [34].

Compound **1** showed [M+H]^+^ and [M+Na]^+^ at *m/z* 343.12331 and 365.10508, and gave [M+Na-C_6_H_10_O_5_]^+^ at *m/z* 203.0519, suggesting the presence of disaccharide structure. Leuine (**2**) and tryptophan (**13**) were two amino acids with typical fragmentation ions by the loss of CO_2_ and NH_3_, and further confirmed by comparison with their reference standards. Compounds **7**, **18,** and **19** were three iridoids, and precisely identified as morroniside, loganin, and geniposide when compared with their reference standards. Guanosine (**6**) and adenosine (**8**) were two nucleosides. Compound **8** was directly identified as adenosine by comparison with its reference standard. Compound **6** was tentatively identified with accurate molecular weight and fragment ions in both positive and negative modes. Compound **11** gave an [M−H]^−^ ion at *m/z* 218.1036 and was tentatively identified as N-(tert-Butoxycarbonyl)threonine. Compounds **62** and **63** were both pentacyclic triterpenoids with [M−H]^−^ ions at m/z 471.34891 and 455.35374, and were tentatively identified as hederagenin and oleanolic acid, respectively [33]. Compound **67** gave an [M−H]^−^ ion at *m/z* 399.32889, calculated to be C_27_H_44_O_2_, and was inferred as dehydrotocopherol. 

It can be shown that the flavonoid and phenolic contents in *P. suffruticosa* flower changes significantly during the flowering development stage (Figure 5). In this study, all compounds were detectable both in BEE and PEE. *p*-Hydroxybenzoic acid (**15**) could not be detected in FEE, and geniposide (**19**) and benzoylpaeoniflorin (**54**) could not be detected in the SEE. On the basis of the peak intensities in negative mode, there were 24 compounds with 10-fold intensity change among the four extracts, including 8 terpenoids (morroniside (**7**), mudanoside A (**12**), mudanpioside E (**23**), galloyloxypaeoniflorin (**30**), mudanploside H (**42**), mudanpioside C (**49**), benzoyloxypaeoniflorin (**51**), and benzoylpaeoniflorin (**54**)), 10 flavonoids (eriodictyol-7-*O*-glucoside (**22**), quercetin-*O*-di-glucoside or isomer (**26, 33**), kaempferol-3,7-di-*O*-glucoside (**29**), patuletin-3,5-di-*O*-glucoside (**32**), quercetin-di-*O*-glucoside or isomer (**33**), 6,3′-dimethoxyquercetin-di-*O*-glucoside (**37**), monoxerutin (**38**), isorhamnetin-3-*O*-glucoside (**44**), apigenin (**55**), and chrysoeriol (**56**)), 4 phenols (gentisic acid-5-*O*-glucoside (**14**), *p*-hydroxybenzoic acid (**15**), 3,4-dimethoxyphenol (**25**), and paeonolide (**27**)), and 2 fatty acids (9,12,13-trihydroxyoctadec-10-enoic acid (**58**) and 3,12-dihydroxyhexadecanoic acid (**59**)). Considering the distribution of these largest abundant compounds, there were 4 compounds (**7**, **12**, **14,** and **26**), one compound (**56**), 9 compounds (**22, 25**, **27**, **29**, **42**, **49**, **51**, **54**, and **55**), and 10 compounds (**15**, **23**, **30**, **32**, **33**, **37**, **38**, **44**, **58**, and **59**) in BEE, FEE, PEE, and SEE, respectively. Obvious variation was easily found in the four extracts from the different flowering stages. The harvest stage was of the highest importance for the application of flowers, which affects chemical composition, nutritional value, and bioactivity. Flower harvesting at the correct stage could significantly increase quality, and even reduce the difficulty of product development.

### 3.7. Determination of Phytochemicals in Ethanolic Extracts of P. suffruticosa Flowers

In this study, though there were 68 compounds found by UFLC-Q-TOF-MS analysis, it was difficult to quantify all the 68 compounds by HPLC-DAD analysis due to the trace contents in the extracts, no/weak UV absorption, and no available standards commercially. Moreover, phenols, monoterpenoids, and flavonoids had multiple bioactivities, and they were predominant constituents in *P. suffruticosa* flowers. The method developed for the simultaneous determination of 3 phenols, 3 monoterpenoids, 2 flavonoids was validated by HPLC-DAD (Table 3). Gallic acid (**5**), 1,2,3,4,6-*O*-pentagalloyl glucose (**35**), and paeonol (**50**) were determined in the four extracts (Table 4). In BEE, the content of 1,2,3,4,6-*O*-pentagalloyl glucose (**35**) was 197.20 ± 6.24 mg/g ext., which was higher than that of gallic acid (**5**) (159.99 ± 5.06 mg/g ext.) and paeonol (**50**) (1.68 ± 0.05 mg/g ext.). Unexpectedly, the content of gallic acid (**5**) was higher than that of 1,2,3,4,6-*O*-pentagalloylglucose (**35**) and paeonol (**50**) in the other three extracts. Oxypaeoniflorin (**21**) (6.93 ± 0.17 mg/g ext.), paeoniflorin (**28**) (16.19 ± 0.40 mg/g ext.), and benzoylpaeoniflorin (**54**) (0.48 ± 0.01 mg/g ext.) were able to be determined in PEE. The content of paeoniflorin (**28**) (19.61 ± 0.38 mg/g ext.) was much higher than that of oxypaeoniflorin (**21**) (10.46 ± 0.20 mg/g ext.) in FEE, but the content of paeoniflorin (**28**) (10.84 ± 0.49 mg/g ext.) was a little lower than that of oxypaeoniflorin (**21**) (11.19 ± 0.50 mg/g ext.) in SEE. Benzoylpaeoniflorin (**54**) was not determined in FEE and SEE. The contents of luteolin (**53**) and apigenin (**55**) in PEE were 1.23 ± 0.03 mg/g ext. and 1.48 ± 0.04 mg/g ext., respectively. The contents of luteolin (**53**) in BEE, FEE, and SEE were lower than 0.16 mg/g ext., approximately one eighth of that in PEE. These results indicated that gallic acid (**5**) and 1,2,3,4,6-*O*-pentagalloyl glucose (**35**) were predominant compounds in ethanolic extracts of *P. suffruticosa* flowers, especially BEE.

### 3.8. Pearson’s Correlation Analysis between Phytochemicals and Bioactivities

Pearson’s correlation analysis has been used for the better understanding of the relationship between the phytochemicals and their bioactivities [35]. The correlation coefficient (r) of a single ingredient to each bioactivity is shown in Figure 6. 

Total phenolic and total flavonoid contents had a high negative correlation with DPPH and hydroxyl radical scavenging. Total phenolic content had a higher correlation with hydroxyl radical scavenging (r = −0.913) rather than DPPH scavenging (r = −0.549). By contrast, total flavonoid content had a higher correlation with DPPH scavenging (r = −0.862). Total phenolic and total flavonoid contents had a positive correlation with the activities of GSH-Px (r_TPC_ = 0.648; r_TFC_ = 0. 960, *p* < 0.05) and SOD (r_TPC_ = 0.885; r_TFC_ = 0.811), and a negative correlation with the contents of TNF-α (r_TPC_ = −0.712; r_TFC_ = −0.847) and IL-6 (r_TPC_ = −0.880; r_TFC_ = −0.783).

The correlation of phytochemicals and the antioxidant activity were further investigated. DPPH scavenging had no significant correlation with any compound. Hydroxyl radical scavenging had a significantly negative correlation with leucine (**2**), 8-debenzoylpaeoniflorin (**3**), gallic acid (**5**), tryptophan (**13**) and trigalloyl glucose isomer (**20**), and a significantly positive correlation with glucogallin or isomer (**9**) and malyngic acid (**57**). The activity of GSH-Px had a significantly positive correlation with geniposide (**19**) and a significantly negative correlation with *p*-hydroxybenzoic acid (**15**) and 3,12-dihydroxyhexadecanoic acid (**59**). The activity of SOD had a significantly positive correlation with gentisic acid-5-*O*-glucoside (**14**), methyl gallate (**16**), paeoniflorin (**28**), and diosmin (**47**), and a negative correlation with 9,12,13-trihydroxyoctadec-10-enoic acid (**58**) and 3,12-dihydroxyhexadecanoic acid (**59**). Gallic acid (**5**) and its derivatives, such as trigalloyl glucose (**20**), are strong natural antioxidants and can scavenge free radicals [36,37]. It has been reported that administration of geniposide (**19**), paeoniflorin (**28**), and diosmin (**47**) can significantly increase the activities of free radical scavenging enzymes such as SOD and GSH-Px [38,39,40].

It was found that the compounds significantly correlated with the content of IL-6 were identical to those of compounds correlated with the activity of SOD, but their correlations were opposite. It has been reported that paeoniflorin (**28**) and diosmin (**47**), with their potential anti-inflammatory capabilities, could significantly down-regulate the mRNA expression levels of IL-6 [39,41]. The content of TNF-α had a significantly positive correlation with glucogallin or isomer (**9**), and had a significantly negative correlation with leucine (**2**), gallic acid (**5**) and tryptophan (**13**). Interestingly, gallic acid (**5**) can inhibit mast cell-derived inflammatory allergic reactions via pro-inflammatory cytokine expression [42].

Gallic acid (**5**), methyl gallate (**16**), geniposide (**19**), trigalloyl glucose (**20**), paeoniflorin (**28**), and diosmin (**47**) were the critical compounds to the antioxidant, anti-photoaging and anti-inflammatory activities of *P. suffruticosa* flowers according to the results of Pearson’s correlation analysis. These compounds should be the biomarkers in the product development and the quality control of *P. suffruticosa* flowers.

## 4. Conclusions

This study showed that *P. suffruticosa* flowers have different antioxidant and anti-photoaging properties at different flowering stages. The phytochemicals were elucidated, and a significant difference was found in terms of abundance. Meanwhile, the results of chemometric analysis proved that the multiple ingredients in *P. suffruticosa* flowers contributed to their antioxidant and anti-photoaging activities. In order to have the excellent antioxidant and anti-photoaging activities and high levels of bioactive phytochemicals, the optimal time to harvest the *P. suffruticosa* flower is before the early flowering stage. Taken together, this study provides valuable evidence that the *P. suffruticosa* flower has a great potential to be developed as functional material in food and health related industries.

## Figures and Tables

**Figure 1 antioxidants-08-00345-f001:**
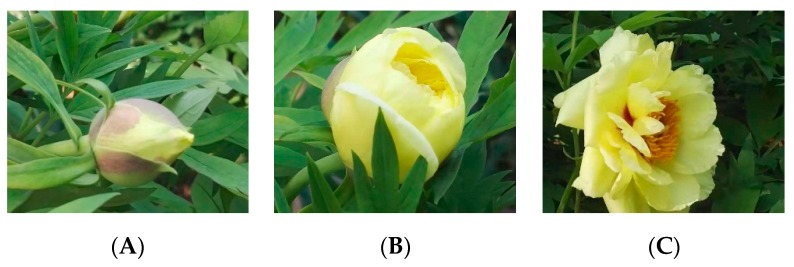
Pictures of *Paeonia suffruticosa* flowers at the different flowering stages. (**A**) *P. suffruticosa* flowering bud; (**B**) *P. suffruticosa* flower at the early flowering stage; (**C**) *P. suffruticosa* flower at the full flowering stage.

**Figure 2 antioxidants-08-00345-f002:**
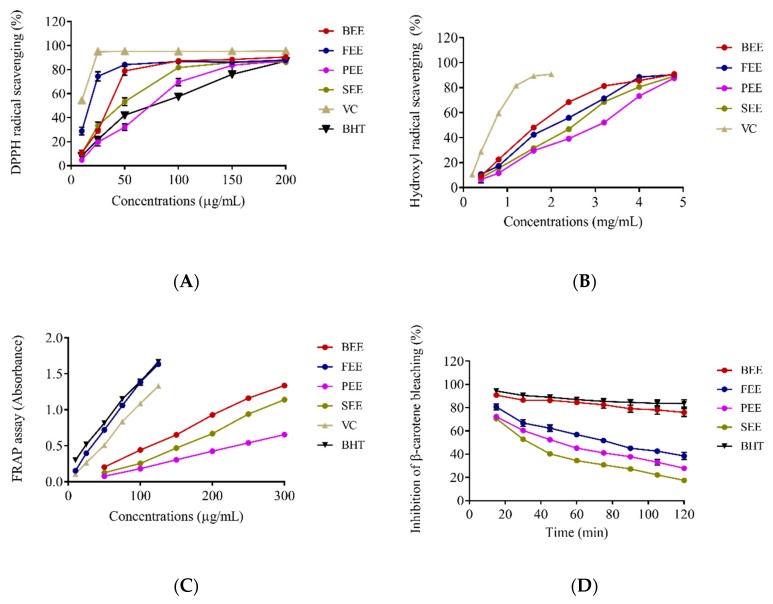
In vitro antioxidant effects of ethanolic extract of *Paeonia*
*suffruticosa* flowers. Effects of ethanolic extract of *P. suffruticosa* flowers on DPPH scavenging (**A**), hydroxyl radical scavenging (**B**), FRAP assay (**C**), and inhibition of β-carotene bleaching (**D**). Data were expressed as mean ± SD (*n* = 3). VC: vitamin C; BHT: butylated hydroxytoluene; BEE: bud ethanolic extract; FEE: flower ethanolic extract; PEE: petal ethanolic extract; SEE: stamen ethanolic extract.

**Figure 3 antioxidants-08-00345-f003:**
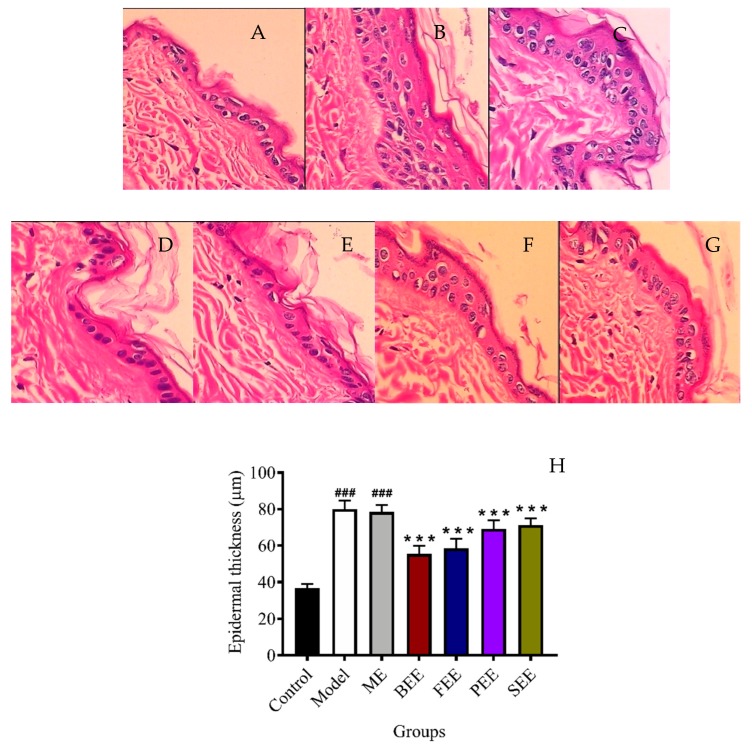
Effects of ethanolic extract of *Paeonia suffruticosa* flowers on UVB-induced epidermal thickening in the mice. Representative images of histological observation by H&E staining of mouse dorsal skin (400×). (**A**) control group: no irradiation; (**B**) model group: irradiated group with no treatment; (**C**) model + emulsifier (ME) group: irradiated group treated with emulsifier; (**D**) bud ethanolic extract (BEE) group: irradiated group treated with 2.0% (*w/w*) BEE in emulsifier; (**E**) flower ethanolic extract (FEE) group: irradiated group treated with 2.0% (*w/w*) FEE in emulsifier; (**F**) petal ethanolic extract (PEE) group: irradiated group treated with 2.0% (*w/w*) PEE in emulsifier; (**G**) stamen ethanolic extract (SEE) group: irradiated group treated with 2.0% (*w/w*) SEE in emulsifier. (**H**) Epidermal thickness. Data were expressed as mean ± SD (*n* = 6). *** *p* < 0.001 versus ME group; ^###^
*p* < 0.001 versus control group.

**Figure 4 antioxidants-08-00345-f004:**
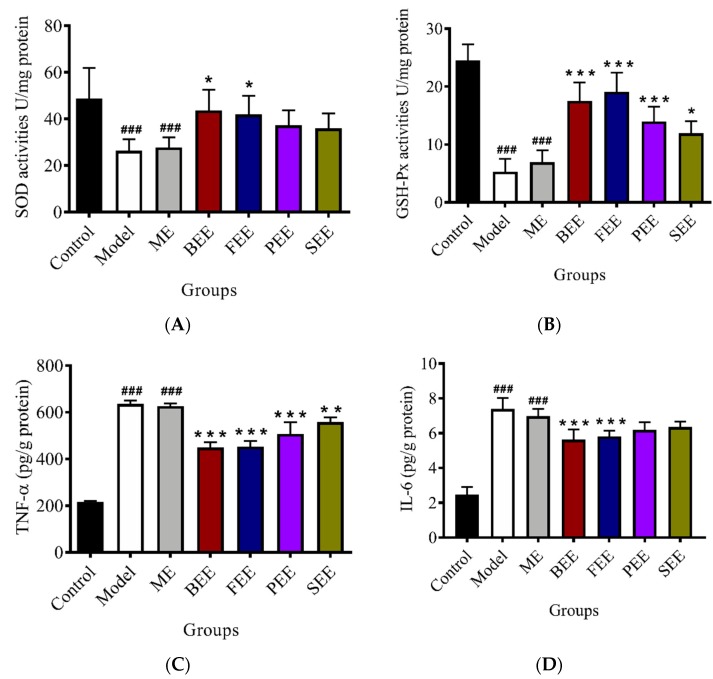
Effects of ethanolic extracts of *Paeonia suffruticosa* flowers on the activities of antioxidant enzymes and the concentrations of pro-inflammatory cytokines in UVB-irradiated mouse skin. Concentrations of SOD (**A**), GSH-Px (**B**), TNF-α (**C**), and IL-6 (**D**) in dorsal skin tissues were determined using mouse-specific enzyme-linked immunosorbent assay (ELISA) kits. Control group: no irradiation; Model group: irradiated group with no treatment; Model + emulsifier (ME) group: irradiated group treated with emulsifier; Bud ethanolic extract (BEE) group: irradiated group treated with 2.0% (*w/w*) BEE in emulsifier; Flower ethanolic extract (FEE) group: irradiated group treated with 2.0% (*w/w*) FEE in emulsifier; Petal ethanolic extract (PEE) group: irradiated group treated with 2.0% (*w/w*) PEE in emulsifier; Stamen ethanolic extract (SEE) group: irradiated group treated with 2.0% (*w/w*) SEE in emulsifier. Data were expressed as mean ± SD (*n* = 6). * *p* < 0.05, ** *p* < 0.01, and *** *p* < 0.001 versus ME group; ^###^
*p* < 0.001 versus control group.

**Figure 5 antioxidants-08-00345-f005:**
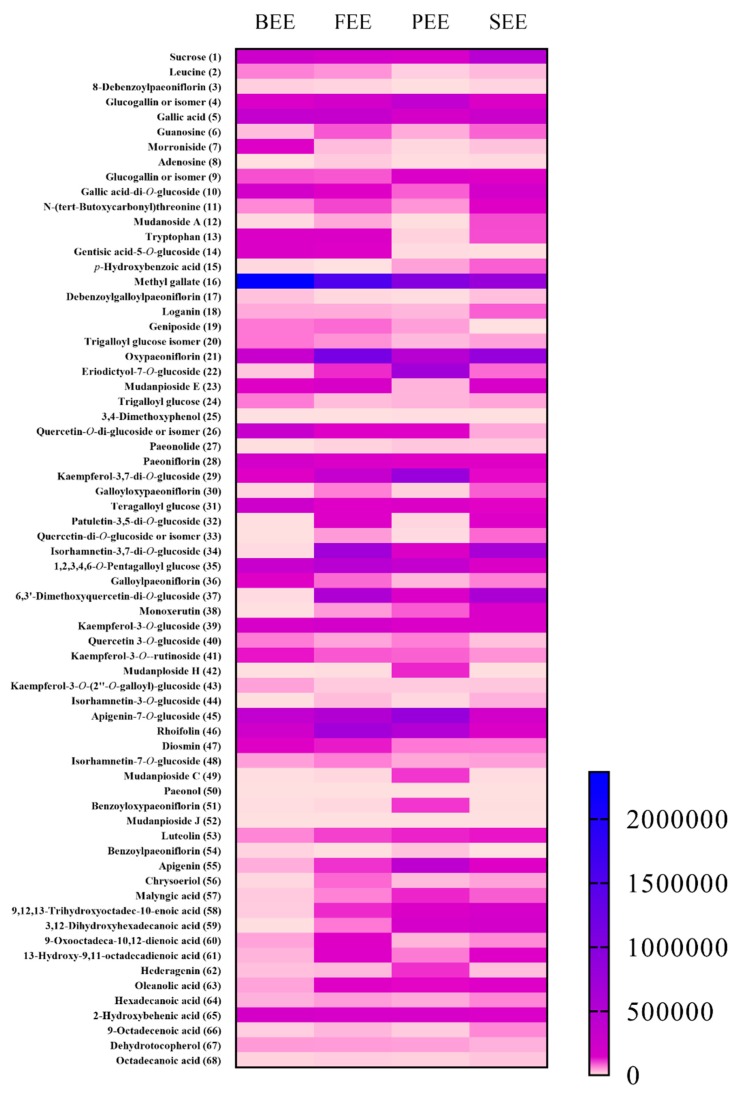
The peak intensities of the ethanolic extracts of *Paeonia suffruticosa* flowers. The peak intensities of the related compound in bud ethanolic extract (BEE), flower ethanolic extract (FEE), petal ethanolic extract (PEE), and stamen ethanolic extract (SEE) were obtained in negative mode. The numbered compounds were consistent with that in Table 2.

**Figure 6 antioxidants-08-00345-f006:**
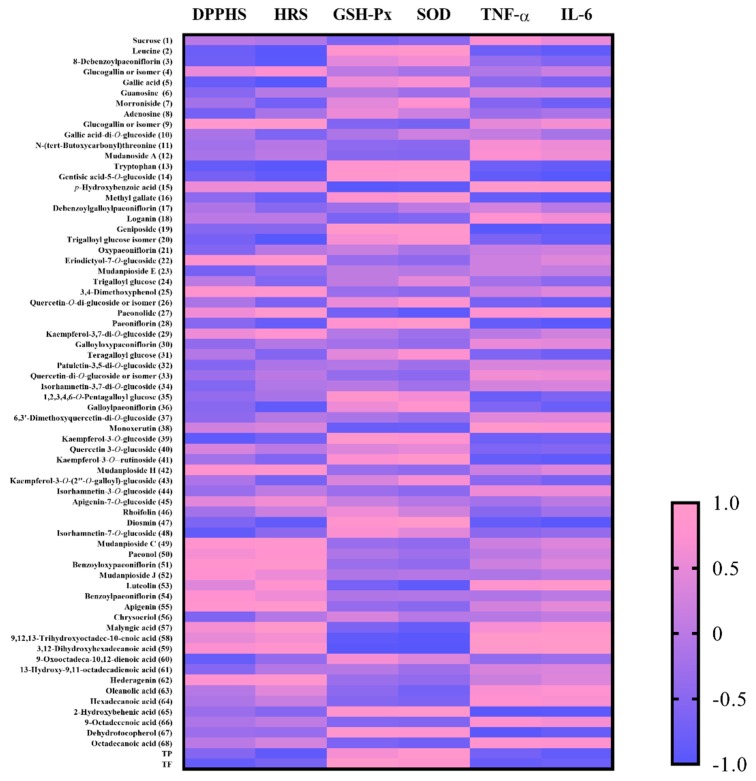
Correlation analysis between phytochemicals and bioactivities. The numbered compounds were consistent with that in Table 2. DPPHS: DPPH scavenging; HRS: hydroxyl radical scavenging.

**Table 1 antioxidants-08-00345-t001:** Extraction yields, total phenolic, and total flavonoid contents, and antioxidant activities (IC_50_ values) of *Paeonia suffruticosa* flowers.

Items	VC	BHT	Samples			
			BEE	FEE	PEE	SEE
Extraction yields (%)	--	--	35.46 ± 3.12	50.76 ± 1.95	42.72 ± 2.48	25.77 ± 4.54
Total phenolic content (mg GAE/g ext.)	--	--	191.80 ± 3.44	113.84 ± 0.84	51.66 ± 0.34	77.25 ± 0.58
Total flavonoid content (mg RE/g ext.)	--	--	30.60 ± 0.94	49.87 ± 0.82	12.98 ± 0.78	8.86 ± 0.98
DPPH radical scavenging activity (μg/mL)	6.16 ± 0.46	64.49 ± 4.68	34.44 ± 1.35	14.83 ± 2.03	64.00 ± 5.60	42.70 ± 0.79
Hydroxyl radical scavenging activity (mg/mL)	0.61 ± 0.02	--	1.57 ± 0.04	1.79 ± 0.02	2.57 ± 0.06	2.10 ± 0.004

VC: vitamin C; BHT: butylated hydroxytoluene; BEE: bud ethanolic extract; FEE: flower ethanolic extract; PEE: petal ethanolic extract; SEE: stamen ethanolic extract. Data were expressed as mean ± SD (*n* = 3).

**Table 2 antioxidants-08-00345-t002:** The compounds identified in the ethanolic extracts of *Paeonia suffruticosa* flowers by UFLC-Q-TOF-MS.

No.	Rt (min)	Molecular Formula	[M+H]^+^ (ppm)	[M−H]^−^ (ppm)	Fragments in Positive Mode	Fragments in Negative Mode	Identification
**1**	1.03	C_12_H_22_O_11_	343.12331 (−0.5)	341.10911 (0.5)	365.10508 [M+Na]^+^, 203.0519 [M+Na−C_6_H_10_O_5_]^+^	179.0557 [M−H−C_6_H_10_O_5_]^−^	Sucrose
**2 ***	1.69	C_6_H_13_NO_2_	132.1018 (−0.8)	130.08735 (0.02)	86.0986 [M+H−HCOOH]^+^, 69.0731 [M+H−HCOOH-NH_3_]^+^	112.0398 [M−H−H_2_O]^−^, 86.0338 [M−H−CO_2_]^−^	Leucine
**3**	1.77	C_16_H_24_O_10_	377.14221 (−5.3)	375.12986 (0.5)	377.14293 [M+NH_4_]^+^, 197.0778 [M+H−C_6_H_10_O_5_−H_2_O]^+^, 179.0 [M+H−C_6_H_10_O_5_−2H_2_O]^+^	195.0581 [M−H−C_6_H_10_O_5_−H_2_O]^−^	8-Debenzoylpaeoniflorin
**4**	2.10	C_13_H_16_O_10_	333.08113 (−1.5)	331.06748 (1.2)	350.10806 [M+NH_4_]^+^	169.0138 [M−H−C_6_H_10_O_5_]^−^, 125.0340 [M−H−C_6_H_10_O_5_−CO_2_]^−^	Glucogallin or isomer
**5 ***	2.30	C_7_H_6_O_5_	171.02846 (−2)	169.01513 (5.2)	153.0181 [M+H−H_2_O]^+^, 127.0392 [M+H−CO_2_]^+^, 109.1294 [M+H−CO_2_−H_2_O]^+^	125.0245 [M−H−CO_2_]^−^	Gallic acid
**6**	2.48	C_10_H_13_N_5_O_5_	284.09891 (−0.1)	282.08447 (0.3)	152.0260 [M+H−C_5_H_8_O_4_]^+^, 135.0293 [M+H−C_5_H_8_O_4_−NH_3_]^+^	150.0419 [M−H−C_5_H_8_O_4_]^−^, 133.0159 [M−H−C_5_H_8_O_4_−NH_3_]^−^	Guanosine
**7 ***	2.70	C_17_H_26_O_11_		405.13989 (−0.8)	424.18247 [M+H+NH_4_]^+^	359.1325 [M−H−HCOOH]^−^, 197.0806 [M−H−HCOOH−C_6_H_10_O_5_]^−^	Morroniside
**8 ***	2.74	C_10_H_13_N_5_O_4_	268.10411 (0.3)	266.08843 (−3.9)	136.0616 [M+H−C_5_H_8_O_4_]^+^, 119.0355 [M+H−C_5_H_8_O_4_−NH_3_]^+^	134.0437 [M−H−C_5_H_8_O_4_]^−^	Adenosine
**9**	3.20	C_13_H_16_O_10_	331.06755 (1.4)	331.06745 (1.1)	315.0683 [M+H−H_2_O]^+^, 153.0172 [M+H−H_2_O−C_6_H_10_O_5_]^+^	241.0325 [M−H−C_3_H_6_O_3_]^−^, 169.0138 [M−H−C_6_H_10_O_5_]^−^, 125.0340 [M−H−C_6_H_10_O_5_−CO_2_]^−^	Glucogallin or isomer
**10**	3.70	C_19_H_26_O_15_	495.13369 (−1.5)	493.12022 (0.7)	333.0772 [M+H−C_6_H_10_O_5_]^+^, 315.0711 [M+H−C_6_H_12_O_6_]^+^, 297.0592 [M+H−C_6_H_12_O_6_−H_2_O]^+^, 171.0272 [M+H−2C_6_H_10_O_5_]^+^, 153.0178 [M+H−C_6_H_12_O_6_−C_6_H_10_O_5_]^+^	313.0546 [M−H−C_6_H_12_O_6_]^−^, 169.0124 [M−H−2C_6_H_10_O_5_]^−^	Gallic acid-di-*O*-glucoside
**11**	4.10	C_9_H_17_NO_5_	220.11782 (−0.6)	218.1036 (0.9)	202.1071 [M+H−H_2_O]^+^, 184.0959 [M+H−2H_2_O]^+^, 174.1126 [M+H−HCOOH]^+^	146.0825 [M−H−C_4_H_10_O]^−^,	N-(tert-Butoxycarbonyl)threonine
**12**	4.60	C_14_H_18_O_9_	331.10195 (−1.2)	329.08812 (0.9)	169.0427 [M+H−C_6_H_10_O_5_]^+^	167.0341 [M−H−C_6_H_10_O_5_]^−^	Mudanoside A
**13 ***	4.86	C_11_H_12_N_2_O_2_	205.09701 (−0.7)	203.08298 (1.9)	188.0709 [M+H−NH_3_]^+^, 118.0657 [M+H−CO_2_−C_2_H_5_N]^+^	159.0911 [M−H−CO_2_]^−^, 116.0513 [M−H−CO_2_−C_2_H_5_N]^−^	Tryptophan
**14**	5.06	C_13_H_16_O_9_	317.08639 (−1)	315.07227 (0.4)	155.0321 [M+H−C_6_H_10_O_5_]^+^, 137.0229 [M+H−C_6_H_10_O_5_−H_2_O]^+^	153.0189 [M−H−C_6_H_10_O_5_]^−^, 109.0303 [M−H−C_6_H_10_O_5_−CO_2_]^−^	Gentisic acid-5-*O*-glucoside
**15 ***	5.91	C_7_H_6_O_3_	139.03894 (−0.3)	137.02588 (10.7)	121.0279 [M+H−H_2_O]^+^, 95.0495 [M+H−CO_2_]^+^, 77.0400 [M+H−CO_2_−H_2_O]^+^	93.0363 [M−H−CO_2_]^−^	*p*-Hydroxybenzoic acid
**16**	6.30	C_8_H_8_O_5_	185.04434 (−0.6)	183.03088 (5.4)	153.0188 [M+H−CH_3_OH]^+^, 125.0239 [M+H−CH_3_COOH]^+^, 107.0137 [M+H−CH_3_COOH−H_2_O]^+^	168.0065 [M−H−CH_3_]^−^, 124.0174 [M−H−CH_3_−CO_2_]^−^	Methyl gallate
**17**	6.58	C_23_H_28_O_14_	529.15249 (−5.1)	527.14061 (0)	493.1146 [M+H−2H_2_O]^+^, 315.0648 [M+H−C_7_H_4_O_5_−2H_2_O]^+^, 179.0679 [C_10_H_11_O_3_]^+^	345.1178 [M−H−C_8_H_6_O_5_]^−^, 313.0594 [M−H−C_7_H_4_O_5_−2H_2_O]^−^, 271.0502, 211.0201, 169.0146 [C_10_H_11_O_3_]^−^	Debenzoylgalloylpaeoniflorin
**18 ***	6.76	C_17_H_26_O_10_	391.16024 (0.9)	389.14563 (0.8)		343.1424 [M−H−HCOOH]^−^, 181.0874 [M−H−HCOOH−C_6_H_10_O_5_]^−^, 163.0764 [M−H−HCOOH−C_6_H_10_O_5_−H_2_O]^−^	Loganin
**19 ***	7.15	C_17_H_24_O_10_		387.12994 (0.7)		341.1357 [M−H−HCOOH]^−^, 179.0705 [M−H−HCOOH−C_6_H_10_O_5_]^−^	Geniposide
**20**	7.28	C_27_H_24_O_18_	637.10441 (1.4)	635.08956 (0.9)	467.0798 [M+H−C_7_H_6_O_3_]^+^, 297.0615 [M+H−2C_7_H_6_O_3_]^+^, 279.0415 [M+H−2C_7_H_6_O_3_−H_2_O]^+^, 153.0171 [C_7_H_5_O_4_]^+^	465.1075 [M−H−C_7_H_6_O_3_]^−^, 295.0506 [M−H−2C_7_H_6_O_3_]^−^, 169.1023 [C_7_H_5_O_5_]^−^	Trigalloyl glucose isomer
**21 ***	7.53	C_23_H_28_O_12_	497.1651 (−0.5)	495.15083 (0.1)	479.1557 [M+H−H_2_O]^+^, 335.1123 [M+H−C_6_H_10_O_5_]^+^, 317.1018 [M+H−H_2_O−C_6_H_10_O_6_]^+^, 197.0810 [M+H−C_6_H_10_O_6_−C_7_H_6_O_3_]^+^, 179.0704 [M+H−C_6_H_10_O_6_−C_7_H_6_O_3_−H_2_O]^+^, 151.0752 [M+H−C_6_H_10_O_6_−C_7_H_6_O3−2H_2_O]^+^, 133.0643 [M+H−C_6_H_10_O_6_−C_7_H_6_O_3_−3H_2_O]^+^	465.1446 [M−H−CH_2_O]^−^, 333.0998 [M−H−C_6_H_10_O_5_]^−^, 195.0657 [M−H−C_6_H_10_O_6_−C_7_H_6_O_3_]^−^, 165.0549 [M−H−C_6_H_10_O_6_−C_7_H_6_O_3_−CH_2_O]^−^, 137.0348 [M−H−C_6_H_10_O_6_−C_7_H_6_O_3_−CH_2_O−H_2_O]^−^	Oxypaeoniflorin
**22**	8.23	C_21_H_22_O_11_	451.12391 (0.9)	449.10937 (1)	289.0705 [M+H−C_6_H_10_O_5_]^+^, 271.0590 [M+H−C_6_H_10_O_5_−H_2_O]^+^, 153.0168	287.0562 [M−H−C_6_H_10_O_5_]^−^, 259.0618, 151.0034	Eriodictyol-7-*O*-glucoside
**23**	8.45	C_24_H_30_O_13_		525.16213 (1.5)	544.202 [M+NH_4_]^+^, 365.1232 [M+H−C_6_H_10_O_5_]^+^, 347.1106 [M+H−C_6_H_10_O_5_−H_2_O]^+^, 197.0797 [M+H−C_6_H_12_O_6_−C_8_H_6_O_3_]^+^, 179.0689 [M+H−C_6_H_12_O_6_−C_8_H_6_O_3_−H_2_O]^+^	495.1558 [M−H−CH_2_O]^−^, 345.1044 [M−H−C_6_H_12_O_6_]^−^, 195.0652 [M−H−C_6_H_12_O_6_−C_8_H_6_O_3_]^−^, 177.0639 [M−H−C_6_H_12_O_6_−C_8_H_6_O_3_−H_2_O]^−^	Mudanpioside E
**24**	8.64	C_27_H_24_O_18_		635.08955 (0.9)	654.12968 [M+NH_4_]^+^, 619.0922 [M+H−H_2_O]^+^, 449.0770 [M+H−H_2_O−C_7_H_6_O_5_]^+^, 297.0615 [M+H−H_2_O−C_7_H_6_O_5_−C_7_H_4_O_4_]^+^, 279.0476 [M+H−2H_2_O−C_7_H_6_O_5_−C_7_H_4_O_4_]^+^	465.0666 [M−H−C_7_H_6_O_5_]^−^, 313.0554 [M−H−C_7_H_6_O_5_−C_7_H_4_O_4_]^−^, 241.05094 [M−H−C_7_H_6_O_5_−2C_7_H_4_O_4_]^−^	Trigalloyl glucose
**25**	8.79	C_8_H_10_O_3_	155.07034 (0.5)		140.0472 [M+H−CH_3_]^+^, 123.0441 [M+H−CH_4_O]^+^		3,4-Dimethoxyphenol
**26**	9.20	C_27_H_30_O_17_	627.15612 (0.9)	625.14123 (0.3)	465.1022 [M+H−C_6_H_10_O_5_]^+^, 303.0497 [M+H−2C_6_H_10_O_5_]^+^	463.0928 [M−H−C_6_H_10_O_5_]^−^, 301.0358 [M−H−2C_6_H_10_O_5_]^−^	Quercetin-*O*-di-glucoside or isomer
**27**	9.67	C_20_H_28_O_12_	461.16556 (0.5)	459.14963 (−2.6)	299.1206 [M+H−C_6_H_10_O_5_]^+^, 167.0707 [M+H−C_6_H_10_O_5_−C_5_H_8_O_4_]^+^	297.0885 [M−H−C_9_H_10_O_3_]^−^, 165.0551 [M−H−C_6_H_10_O_5_−C_5_H_8_O_4_]^−^	Paeonolide
**28 ***	10.03	C_23_H_28_O_11_	481.17026 (−0.4)	479.15559 (−0.6)	319.1260 [M+H−C_6_H_10_O_5_]^+^, 301.1567 [M+H−C_6_H_10_O_5_−H_2_O]^+^, 179.0703 [M+H−C_6_H_10_O_5_−H_2_O−C_7_H_6_O_2_]^+^,	449.1520 [M−H−CH_2_O]^−^, 327.1088 [M−H−C_7_H_6_O_2_]^−^, 165.0551 [M−H−C_7_H_6_O_2_−C_6_H_10_O_5_]^−^	Paeoniflorin
**29**	10.27	C_27_H_30_O_16_	611.16221 (2.5)	609.14676 (1.1)	449.1081 [M+H−C_6_H_10_O_5_]^+^, 287.0552 [M+H−2C_6_H_10_O_5_]^+^	447.0961 [M−H−C_6_H_10_O_5_]^−^, 285.0403 [M−H−2C_6_H_10_O_5_]^−^	Kaempferol-3,7-di-*O*-glucoside
**30**	10.48	C_30_H_32_O_16_	649.17332 (−4.6)	647.1606 (−1.8)	511.1395 [M+H−C_7_H_6_O_3_]^+^, 315.0712 [M+H−C_17_H_18_O_7_]^+^, 153.0174 [M+H−C_17_H_18_O_7_−C_6_H_10_O_5_]^+^	509.1335 [M−H−C_7_H_6_O_3_]^−^, 313.0559 [M−H−C_17_H_18_O_7_]^−^,	Galloyloxypaeoniflorin
**31**	10.78	C_34_H_28_O_22_	789.11272 (−2.3)	787.10067 (0.9)	771.1010 [M+H−H_2_O]^+^, 619.0877 [M+H−C_7_H_4_O_4_−H_2_O]^+^, 449.0694 [M+H−2C_7_H_4_O_4_−2H_2_O]^+^, 279.0476 [M+H−3C_7_H_4_O_4_−3H_2_O]^+^	635.0985 [M−H−C_7_H_4_O_4_]^−^, 617.0932 [M−H−C_7_H_4_O_4_−H_2_O]^−^, 465.0734 [M−H−2C_7_H_4_O_4_−H_2_O]^−^, 295.0473 [M−H−3C_7_H_4_O_4_−2H_2_O]^−^,	Teragalloyl glucose
**32**	11.28	C_28_H_32_O_18_	657.16586 (−0.4)	655.135 (2.9)	495.118 [M+H−C_6_H_10_O_5_]^+^, 333.0606 [M+H−2C_6_H_10_O_5_]^+^	331.0490 [M−H−2C_6_H_10_O_5_]−,	Patuletin-3,5-di-*O*-glucoside
**33**	11.69	C_27_H_30_O_17_	627.1546 (−1.6)	625.1419 (1.4)	465.1038 [M+H− C_6_H_10_O_5_]^+^, 303.0500 [M+H−2 C_6_H_10_O_5_]^+^	301.0337 [M−H− C_6_H_10_O_5_]^−^, 271.0245 [M−H−2 C_6_H_10_O_5_−CH_2_O]^−^	Quercetin-di-*O*-glucoside or isomer
**34**	12.21	C_28_H_32_O_17_	641.17086 (−0.6)	639.15895 (3.6)	479.1168 [M+H−C_6_H_10_O_5_]^+^, 317.0652 [M+H−2C_6_H_10_O_5_]^+^	315.0507 [M−H−2C_6_H_10_O_5_]^−^	Isorhamnetin-3,7-di-*O*-glucoside
**35 ***	12.33	C_41_H_32_O_26_	941.12626 (0.8)	939.11461 (3.9)	771.110 [M+H−C_7_H_6_O_5_]^+^, 431.0623 [M+H−3C_7_H_6_O_5_]^+^, 279.0462 [M+H−3C_7_H_6_O_5_−C_6_H_10_O_5_]^+^,	769.1070 [M−H−C_7_H_6_O_5_]^−^, 617.0931 [M−H− C_7_H_6_O_5_−C_7_H_4_O_4_]^−^, 465.07051 [M−H− C_7_H_6_O_5_−2C_7_H_4_O_4_]^−^, 295.0430 [M−H−2 C_7_H_6_O_5_−2C_7_H_4_O_4_]^−^	1,2,3,4,6-*O*-Pentagalloyl glucose
**36**	12.36	C_30_H_32_O_15_	633.18077 (−1)	631.16737 (0.8)	315.0721 [M+H−C_7_H_6_O_2_−C_10_H_12_O_10_]^+^, 179.0689 [C_10_H_11_O_3_]^+^	613.1706 [M−H−H_2_O]^−^, 509.1421 [M−H− C_7_H_6_O_2_]^−^, 313.1568 [M−H− C_7_H_6_O_2_− C_10_H_12_O_10_]^−^, 271.0539 [M−H− C_7_H_6_O_2_− C_10_H_12_O_10_−C_2_H_2_O]^−^	Galloylpaeoniflorin
**37**	12.47	C_29_H_34_O_18_	671.18123 (−0.8)	669.16952 (3.4)	509.1271 [M+H−C_6_H_10_O_5_]^+^, 347.0753 [M+H−2C_6_H_10_O_5_]^+^,	345.0623 [M−H−2C_6_H_10_O_5_]^−^, 301.0361 [M−H−2C_6_H_10_O_5_−CO_2_]^−^	6,3′-Dimethoxyquercetin-di-*O*-glucoside
**38**	12.98	C_29_H_34_O_17_	655.18661 (−0.4)	653.17535 (4.6)	509.1293 [M+H−C_5_H_8_O_4_]^+^, 347.0763 [M+H−C_5_H_8_O_4_−C_6_H_10_O_5_]^+^	345.0552 [M−H−C_5_H_8_O_4_−C_6_H_10_O_5_]^−^, 301.0335 [M−H−C_5_H_8_O_4_−C_6_H_10_O_5_−C_2_H_4_O]^−^	Monoxerutin
**39**	13.11	C_21_H_20_O_11_	449.10826 ()	447.09399 (1.6)	287.0555 [M+H−C_6_H_10_O_5_]^+^, 153.0181	285 [M−H−C_6_H_10_O_5_]^−^	Kaempferol-3-*O*-glucoside
**40 ***	13.48	C_21_H_20_O_12_		463.08881 (1.3)		301.0354 [M−H−C_6_H_10_O_5_]^−^, 151.0026	Quercetin-3-*O*-glucoside
**41**	13.59	C_27_H_30_O_15_	595.16617 (0.7)	593.15348 (3.9)	287.0549 [M+H−C_12_H_20_O_9_]^+^	285.0425 [M−H−C_12_H_20_O_9_]^−^	Kaempferol-3-*O*-rutinoside
**42**	13.66	C_30_H_32_O_14_	617.18694 (0.7)	615.17285 (1.5)	599.1479 [M+H−H_2_O]^+^, 479.1685 [M+H−C_7_H_6_O_3_]^+^, 443.1294 [M+H−C_7_H_6_O_3_−2H_2_O]^+^, 317.1017 [M+H−C_7_H_6_O_3_−C_6_H_10_O_5_]^+^, 179.0703 [C_10_H_11_O_3_]^+^	431.1396 [M−H−CH_2_O_2_−C_7_H_6_O_3_]^−^, 281.06836 [M−H−CH_2_O_2_−C_7_H_6_O_3_−C_7_H_6_O_2_−CH_2_O]^−^	Mudanploside H
**43**	13.94	C_28_H_24_O_15_	601.11847 (−0.5)	599.10555 (2.2)	287.0544 [M+H−C_13_H_14_O_9_]^+^	285.0413 [M−H−C_13_H_14_O_9_]^−^	Kaempferol-3-*O*- (2′’-*O*-galloyl)-glucoside
**44**	14.28	C_22_H_22_O_12_	479.1188 (0.8)	477.10532 (3.1)	317.0661 [M+H−C_6_H_10_O_5_]^+^	315.0537 [M−H− C_6_H_10_O_5_]^−^, 299.0214 [M−H−C_6_H_10_O_6_]^−^, 271.0272 [M−H−C_6_H_10_O_6_−H_2_O]^−^, 255.0335, 199.0371, 171.0587	Isorhamnetin-3-*O*-glucoside
**45**	14.31	C_21_H_20_O_10_	433.11321 (0.7)	431.10019 (4.2)	271.0600 [M+H−C_6_H_10_O_5_]^+^, 153.0176	269.0449 [M−H−C_6_H_10_O_5_]^−^	Apigenin-7-*O*-glucoside
**46 ***	14.61	C_27_H_30_O_14_	579.1711 (0.5)	577.15834 (3.6)	433.1131 [M+H−C_5_H_8_O_4_]^+^, 271.0606 [M+H−C_5_H_8_O_4_−C_6_H_10_O_5_]^+^, 153.0177	269.0462 [M+H−C_5_H_8_O_4_−C_6_H_10_O_5_]7^−^	Rhoifolin
**47**	14.91	C_28_H_32_O_15_	0.5	607.17007 (5.3)	463.1220 [M+H−C_6_H_10_O_4_]^+^, 301.0703 [M+H−C_6_H_10_O_4_−C_6_H_10_O_5_]^+^, 286.0461 [M+H−C_6_H_10_O_4_−C_6_H_10_O_5_−CH_3_]^+^	443.0939 [M−H−C_6_H_12_O_5_]^−^, 299.0546 [M−H−C_6_H_10_O_4_−C_6_H_10_O_5_]^−^, 284.0301 [M−H−C_5_H_8_O_4_−C_6_H_10_O_5_−CH_3_]^−^	Diosmin
**48**	14.95	C_22_H_22_O_12_	479.11832 (−0.2)	477.10464 (1.7)	317.0668 [M+H−C_6_H_10_O_5_]^+^	285.0372 [M−H−C_6_H_10_O_5_−CH_2_O]^−^	Isorhamnetin-7-*O*-glucoside
**49**	15.34	C_30_H_32_O_13_	601.18754 (−6.7)	599.17976 (4.6)	461.1447 [M+H−C_7_H_6_O_2_−H_2_O]^+^, 443.1269 [M+H−C_7_H_6_O_2_−2H_2_O]^+^, 301.1056 [M+H−C_7_H_4_O_2_−C_7_H_4_O_3_−CO_2_]^+^, 283.08136 [M+H−C_7_H_4_O_2_−C_7_H_4_O_3_−CO_2_−H_2_O]^+^, 179.0695 [C_10_H_11_O_3_]^+^	447.1527 [M−H−CH_2_O−C_7_H_6_O_2_]^−^, 431.1379 [M−H−CH_2_O−C_7_H_6_O_3_]^−^, 281.0682 [M−H−C_7_H_4_O_2_−C_7_H_4_O_3_−CO_2_−H_2_O]^−^, 179.0329 [C_10_H_11_O_3_]^−^	Mudanpioside C
**50 ***	15.69	C_9_H_10_O_3_	167.06995 (−1.9)	165.05663 (5.5)	149.0737 [M+H−H_2_O]^+^, 121.0616 [M+H−H_2_O−CO]^+^	150.0318 [M−H−CH_3_]^−^, 135.0098 [M−H−CH_2_O]^−^, 122.0377 [M−H−CH_3_−CO]^−^	Paeonol
**51**	15.78	C_30_H_32_O_13_		599.18003 (5)		569.1748 [M−H−CH_2_O]^−^, 447.1519 [M−H−CH_2_O−C_7_H_6_O_2_]^−^	Benzoyloxypaeoniflorin
**52**	16.14	C_31_H_34_O_14_	631.20136 (−1.2)	629.19181 (6.7)	457.1797 [M+H−C_8_H_6_O_4_−H_2_O]^+^, 317.1009 [M+H−C_7_H_6_O_2_−C_8_H_6_O_4_−H_2_O]^+^, 297.1009 [M+H−C_7_H_6_O_2_−C_8_H_6_O_4_−2H_2_O]^+^, 279.1108 [M+H−C_7_H_6_O_2_−C_8_H_6_O_4_−3H_2_O]^+^, 179.0692 [C_10_H_11_O_3_]^+^	477.1511 [M−H−C_8_H_6_O_3_]^−^, 333.1206 [M−H−C_7_H_6_O_2_−C_8_H_6_O_4_]^−^	Mudanpioside J
**53 ***	16.95	C_15_H_10_O_6_		285.04116 (2.4)	175.0407, 133.0318	151.0031, 133.0278	Luteolin
**54 ***	17.47	C_30_H_32_O_12_	585.19501 (−2.8)	583.1816 (−0.9)	445.1496 [M+H−C_7_H_6_O_2_−H_2_O]^+^, 427.1407 [M+H−C_7_H_6_O_2_−2H_2_O]^+^, 179.0698 [C_10_H_11_O_3_]^+^	461.1775 [M−H−C_7_H_6_O_2_]^−^, 343.1562 [M−H−C_7_H_4_O_2_−C_7_H_4_O_3_]^−^	Benzoylpaeoniflorin
**55 ***	18.27	C_15_H_10_O_5_		269.0465 (3.5)		225.0553, 149.0235, 117.0346	Apigenin
**56**	18.49	C_16_H_12_O_6_	301.07102 (1.2)	299.05704 (3.1)	286.0479 [M+H−CH_3_]^+^, 258.0531 [M+H−CH_3_−CO]^+^ 153.0158	284.0352 [M−H−CH_3_]^−^, 256.0408 [M−H−CH_3_−CO]^−^	Chrysoeriol
**57**	19.66	C_18_H_32_O_5_	329.23227 (0.1)	327.21889 (3.7)	293.1959 [M+H−2H_2_O]^+^, 275.1959 [M+H−2H_2_O]^+^, 225.1488, 185.1171, 161.1375, 119.0830, 105.0700,	309.2062 [M−H−H_2_O]^−^, 291,1961 [M−H−2H_2_O]^−^, 229.1448, 211.1341, 171.1038	Malyngic acid
**58**	20.57	C_18_H_34_O_5_	331.2482 (0.9)	329.23479 (4.4)	313.2713 [M+H−H_2_O]^+^, 295.2205 [M+H−2H_2_O]^+^, 277.2121 [M+H−3H_2_O]^+^, 213.1447, 195.1343, 171.1285	311.1223 [M−H−H_2_O]^−^, 229.1447, 211.1327, 171.023	9,12,13-Trihydroxyoctadec-10-enoic acid
**59**	21.15	C_16_H_32_O_4_	289.23763 (1.0)	287.22468 (6.6)	271.2277 [M+H−H_2_O]^+^, 253.2155 [M+H−2H_2_O]^+^, 235.2064 [M+H−3H_2_O]^+^, 217.1897 [M+H−4H_2_O]^+^, 161.1364, 135.1159, 111.1162	269.2127 [M−H−H_2_O]^−^, 241.2167 [M−H−HCOOH]^−^	3,12-Dihydroxyhexadecanoic acid
**60**	24.79	C_18_H_30_O_3_	295.22674 (−0.1)	293.21325 (3.5)	277.2145 [M+H−H_2_O]^+^, 231.1648 [M+H−3H_2_O]^+^, 207.1426, 171.1093, 147.1158	275.2024 [M−H−H_2_O]^−^, 223.134, 195.1395	9-Oxooctadeca-10,12-dienoic acid
**61**	25.57	C_18_H_32_O_3_	297.24039 (−6.8)	295.22859 (2.4)	281.0520 [M+H−H_2_O]^+^, 191.0009,133.0103	277.21992 [M−H−H_2_O]^−^, 195.1402	13-Hydroxy-9,11-octadecadienoic acid
**62**	26.63	C_30_H_48_O_4_	473.36163 (−1.9)	471.34891 (2)	427.3607 [M+H−HCOOH]^+^	425.2233 [M−H−HCOOH]^−^	Hederagenin
**63**	28.34	C_30_H_48_O_3_	457.36722 (−0.9)	455.35374 (1.5)	439.3579 [M+H−H_2_O]^+^, 411.3613 [M+H−HCOOH]^+^, 393.3485 [M+H−HCOOH−H_2_O]^+^	409.2504 [M−H−HCOOH]^−^	Oleanolic acid
**64 ***	29.21	C_16_H_32_O_2_	257.24755 (0.2)	255.23339 (1.7)	229.2057 [M+H−CO]^+^	237.1230 [M−H−H_2_O]^−^	Hexadecanoic acid
**65**	29.42	C_22_H_44_O_3_	357.33656 (0.7)	355.32323 (4.1)	339.3238 [M+H−H_2_O]^+^, 321.3230 [M+H−2H_2_O]^+^, 303.3031 [M+H−3H_2_O]^+^	309.3120 [M−H−HCOOH]^−^	2-Hydroxybehenic acid
**66 ***	29.50	C_18_H_34_O_2_	283.26342 (0.9)	281.24868 (0.3)	265.2497 [M+H−H_2_O]^+^, 247.2429 [M+H−2H_2_O]^+^, 237.5873 [M+H−HCOOH]^+^	263.0356 [M−H−H_2_O]^−^	9-Octadecenoic acid
**67**	29.38	C_27_H_44_O_2_	401.34101 (−1)	399.32889 (5.1)	283.3072 [M+H−H_2_O]^+^	355.3114 [M−H−CO_2_]^−^	Dehydrotocopherol
**68 ***	29.65	C_18_H_36_O_2_		283.26451 (0.9)		265.2659 [M−H−H_2_O]^−^	Octadecanoic acid

* The compound was identified by the comparison with reference standard.

**Table 3 antioxidants-08-00345-t003:** Method validation of three phenols, three monoterpenoids, and two flavonoids.

Compounds	Regression Equation	Correlation Coefficient (r)	Linear Range (μg/mL)	LOD (μg/mL)	LOQ (μg/mL)	Intraday Precision (RSD%, n = 6)	Interday Precision (RSD%, *n* = 6)	Accuracy (%, *n* = 6)
Gallic acid (**5**)	y = 9579.15x − 281.13	0.99938	1.61–807.0	0.48	1.61	1.77	2.60	97.05 ± 4.64
Oxypaeoniflorin (**21**)	y = 17106.22x − 36.29	0.99952	1.44–72.00	0.43	1.44	1.96	5.42	97.79 ± 4.90
Paeoniflorin (**28**)	y = 2380.45x − 11.64	0.99987	0.98–246.0	0.29	0.98	1.67	3.24	98.02 ± 4.01
1,2,3,4,6-*O*-Pentagalloyl glucose (**35**)	y = 10624.06x + 58.48	0.99957	1.06–532.0	0.32	1.06	2.57	4.22	96.86 ± 2.27
Luteolin (**53**)	y = 20195.32x − 2.47	0.99999	0.76–19.00	0.23	0.76	1.19	3.93	99.14 ± 3.96
Apigenin (**55**)	y = 17107.34x + 12.70	0.99987	1.72–43.00	0.52	1.72	1.72	2.39	101.5 ± 3.17
Benzoylpaeoniflorin (**54**)	y = 2765.39x − 0.54	0.99999	1.56–39.00	0.47	1.56	1.70	3.50	101.8 ± 5.17
Paeonol (**50**)	y = 17767.49x − 3.55	0.99978	1.96–49.00	0.59	1.96	4.39	3.63	97.74 ± 1.58

**Table 4 antioxidants-08-00345-t004:** Contents of three phenols, three monoterpenoids, and two flavonoids in the bud, flower, petal, and stamen ethanolic extracts of *Paeonia suffruticosa* flowers, as determined using HPLC-DAD analysis.

Compounds	BEE (mg/g ext.)	FEE (mg/g ext.)	PEE (mg/g ext.)	SEE (mg/g ext.)
Gallic acid (**5**)	159.99 ± 5.06	46.98 ± 0.92	32.31 ± 0.80	40.39 ± 1.82
Oxypaeoniflorin (**21**)	nd	10.46 ± 0.20	6.93 ± 0.17	11.19 ± 0.50
Paeoniflorin (**28**)	1.76 ± 0.06	19.61 ± 0.38	16.19 ± 0.40	10.84 ± 0.49
1,2,3,4,6-*O*-Pentagalloyl glucose (**35**)	197.20 ± 6.24	38.72 ± 0.76	26.25 ± 0.65	21.44 ± 0.97
Luteolin (**53**)	0.14 ± 0.004	0.16 ± 0.003	1.23 ± 0.03	0.14 ± 0.01
Apigenin (**55**)	nd	nd	1.48 ± 0.04	nd
Benzoylpaeoniflorin (**54**)	1.30 ± 0.04	nd	0.48 ± 0.01	nd
Paeonol (**50**)	1.68 ± 0.05	1.12 ± 0.02	2.60 ± 0.06	0.36 ± 0.02

BEE: bud ethanolic extract; FEE: flower ethanolic extract; PEE: petal ethanolic extract; SEE: stamen ethanolic extract. nd: not detectable. Data were expressed as mean ± SD (*n* = 3).

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
