# Peer review of "Comparison of Chemical Compositions, Antioxidant, and Anti-Photoaging Activities of Paeonia suffruticosa Flowers at Different Flowering Stages"

_antioxidants, 2019, doi:10.3390/antiox8090345_

Round 1

Reviewer 1 Report

The presented manuscript reports on the ethanolic extracts of P. suffruticosa flowers and their bioactivity depending on the extent of flowering with intention of defining the optimal time for harvesting. The introduction gives the necessary background and is enriched with relevant citations giving an overall satisfying short and compact entry point for the subject. The subject has only been partly addressed making it worthy for investigations; the experimental approach is straight forward, well planned, clearly presented as reflected by the focused objectives. Some methods need short specifications. The results are shortly and well presented. The discussion is also enhanced by a few appropriate comparative examples/citations to scientifically round up the manuscript. Altogether a well performed study.

Specific remarks:

Line 56: The study on the relationship between phytochemicals and bioactivity during the flowering stages is less documented, which is ….?

Line 89: … was repeated after filtration.

Line 93: Please indicate the method used (Folin Ciocalteau procedure?)

Line 99: Please again indicate very shortly the method applied; the reader can always refer to the citation provided, but naming the procedure simplifies the matter, since not very publication is always available.

Line 104: Please elaborate the abbreviation “FRAP” while using it for the first time – this applies also generally to other used abbreviations.

Line 105: Please just indicate shortly the method applied

Line 153: The values for the collision energy 35 eV and collision energy spread 15 eV; are supposed to be in “eV”.

Line 186: Unexpectedly, FEE had the highest … ? (“Unlikely” does not fit well)

Line 207: Unexpectedly, the IC50 Value …..

Line 224: The skin exposed excessively….

Line 289: … and thereby partly compensate the inflammation…

Lines 294-5: … fragmentation pathways, which were carried out with the same kind of …

Line 298: Monoterpenoids are present as ubiquitous chemical compounds in P. suffruticosa, which are characterized by two isoprene units …

Line 321-2: Flavonoids, especially quercetin, kaempferol, isorhamnetin, apigenin and their derivatives, were another set of components also identified in P. suffruticosa

Line 324-5:and thereafter followed by Ret-Diels-Alder (RDA) dissociation, subsequently generating …

Line 385-6: It can be shown that the flavonoid and phenolic contents in P. suffruticosa flower changed significantly during the flowering development stage

Line 410: Unexpectedly, the content ….

Line 455: It was found that the compounds…

Line 466: … and the quality …

Figure 3: Legend - …. Line 259: … the concentrations of pro-inflammatory cytokines

Author Response

Reviewer 1

Q1: Line 56: The study on the relationship between phytochemicals and bioactivity during the flowering stages is less documented, which is ….?

We have revised in lines 56-57.

Q2: Line 89: … was repeated after filtration.

We have revised in line 96.

Q3: Line 93: Please indicate the method used (Folin Ciocalteau procedure?)

We have added the procedure (Part 2.4) in lines 99-106.

Q4: Line 99: Please again indicate very shortly the method applied; the reader can always refer to the citation provided, but naming the procedure simplifies the matter, since not very publication is always available.

We have added the procedure (Part 2.5) in lines 107-115.

Q5: Line 104: Please elaborate the abbreviation “FRAP” while using it for the first time – this applies also generally to other used abbreviations.

We have revised in line 138.

Q6: Line 105: Please just indicate shortly the method applied

We have described the procedure of antioxidant assay in the manuscript (lines 116-161).

Q7: Line 153: The values for the collision energy 35 eV and collision energy spread 15 eV; are supposed to be in “eV”.

We have revised in lines 210-211.

Q8: Line 186: Unexpectedly, FEE had the highest … ? (“Unlikely” does not fit well)

We have revised in line 243.

Q9: Line 207: Unexpectedly, the IC50 Value …..

We have revised in line 264.

Q10: Line 224: The skin exposed excessively….

We have revised in line 282.

Q11: Line 289: … and thereby partly compensate the inflammation…

We have revised in line 363.

Q12: Lines 294-5: … fragmentation pathways, which were carried out with the same kind of …

We have revised in lines 37-374.

Q13: Line 298: Monoterpenoids are present as ubiquitous chemical compounds in P. suffruticosa, which are characterized by two isoprene units …

We have revised in lines 377-378.

Q14: Line 321-2: Flavonoids, especially quercetin, kaempferol, isorhamnetin, apigenin and their derivatives, were another set of components also identified in P. suffruticosa

We have revised in lines 400-401.

Q15: Line 324-5: … and thereafter followed by Ret-Diels-Alder (RDA) dissociation, subsequently generating …

We have revised in line 403.

Q16: Line 385-6: It can be shown that the flavonoid and phenolic contents in P. suffruticosa flower changed significantly during the flowering development stage

We have revised in lines 464-465.

Q17: Line 410: Unexpectedly, the content ….

We have revised in line 494.

Q18: Line 455: It was found that the compounds…

We have revised in line 538.

Q19: Line 466: … and the quality …

We have revised in line 549.

Q20: Figure 3: Legend - …. Line 259: … the concentrations of pro-inflammatory cytokines…

We have revised in line 325.

Reviewer 2 Report

Manuscript by Jingyu et al entitled ‘Comparison of chemical compositions, antioxidant and anti-photoaging activities of Paeonia suffruticosa flowers at different flowering stages’ presents both in vitro and in vivo studies. A lot of effort was made to identify active compounds in the plants. The manuscript is well written and interesting. The manuscript could be accepted for publication after correction of minor issues:

Author should indicate more details regarding staining procedure of dorsal skin tissue of tested animals (In’ Materials and Methods’ section: ‘The immunohistochemical examination was performed according to the manufacturer’s instruction (GTVisionTMI Detection System, Gene Tech, Shanghai, China).’ In Figure 2: ‘Representative images of histological observation by H&E staining of mouse dorsal skin (400 x).’) In ‘Materials and Methods’ section Authors should exactly indicate reference compounds used in appropriate assays and Authors should also briefly describe all antioxidant assays including control and reference experiments. In Table 1 Authors should indicate how data are presented (SD) and should include the data of reference compounds for DPPH and hydroxyl radical assays. In Figure 3 Authors should explain control and model. Compounds numbers should be consequently reported in bold style.

Author Response

Reviewer 2

Q1: Author should indicate more details regarding staining procedure of dorsal skin tissue of tested animals (In’ Materials and Methods’ section: ‘The immunohistochemical examination was performed according to the manufacturer’s instruction (GTVisionTMI Detection System, Gene Tech, Shanghai, China).’ In Figure 2: ‘Representative images of histological observation by H&E staining of mouse dorsal skin (400 x).’)

In this study, we only did H&E staining of dorsal mouse skin. We have revised Part 2.8 Histological analysis in the manuscript (Lines 182-189).

Q2: In ‘Materials and Methods’ section Authors should exactly indicate reference compounds used in appropriate assays and Authors should also briefly describe all antioxidant assays including control and reference experiments.

We have described the procedure of antioxidant assay including control and reference experiments in the manuscript (Lines 116-161).

Q3: In Table 1 Authors should indicate how data are presented (SD) and should include the data of reference compounds for DPPH and hydroxyl radical assays.

We have added the data of reference compounds for DPPH and hydroxyl radical assays in Table 1 and indicated how data are presented (Line 252).

Q4: In Figure 3 Authors should explain control and model.

We have explained the control and model groups in Figure 4 of the manuscript (Lines 327-333).

Q5: Compounds numbers should be consequently reported in bold style.

We have revised in the manuscript (Lines 396-420).

Reviewer 3 Report

The study looked into the antioxidant activity in vitro and anti-photoaging activity in vivo of ethanolic extracts from P. suffruticosa bud, flower at the early flowering stage and the petal and stamen obtained from full flowering stage, then characterized the phytochemical composition change during flowering stage and investigate the relationship between these phytochemicals and their bioactivities using Pearson’s correlation analysis. In conclusion, the authors suggested that the optimal time to harvest P. suffruticosa flower was before the early flowering stage because P. suffruticosa flower have the outstanding antioxidant and anti-photoaging activities in bud and early flowering stage. Some suggestions on this manuscript listed below.

The authors should present the pictures of P. suffruticosa bud, early flowering stage and full flowering stage in this manuscript.

2 I suggested that the authors should measure the epidermal thickness of mice skin in Figure 2 and chart a bar graph.

I suggested that the authors should introduce clearly what is the mechanism of increasing epidermal thickness after UVB-irradiation and how the antioxidant and anti-inflammation properties of the ethanolic extracts ameliorate this pathological symptom in the section 3.3. The authors should explain why they only chose Gallic acid (5), Oxypaeoniflorin (21), Paeoniflorin (28), 1,2,3,4,6-O-Pentagalloyl glucose (35), Luteolin (53), Apigenin (55), Benzoylpaeoniflorin (54) and Paeonol (50) to quantify their contents in four ethanolic extracts using HPLC-DAD analysis but not quantify all the 64 compounds. The animal experiment did not contain a positive control group, which made the quality of this data reduced.

Author Response

Reviewer 3

Q1: The authors should present the pictures of P. suffruticosa bud, early flowering stage and full flowering stage in this manuscript.

We have added the pictures of P. suffruticosa bud, early flowering stage and full flowering stage in Figure 1.

Q2: I suggested that the authors should measure the epidermal thickness of mice skin in Figure 2 and chart a bar graph.

We have added the result of epidermal thickness of mouse skin in the manuscript (Lines 286-291) and a bar graph in Figure 3.

Q3: I suggested that the authors should introduce clearly what is the mechanism of increasing epidermal thickness after UVB-irradiation and how the antioxidant and anti-inflammation properties of the ethanolic extracts ameliorate this pathological symptom in the section 3.3.

We have revised the sentences in lines 282-285 to introduce what is the mechanism of increasing epidermal thickness after UVB-irradiation. In addition, to explain how the antioxidant and anti-inflammation properties of the ethanolic extracts ameliorate epidermal thickening, we have revised the sentences in lines 294-298, 341-346 and 354-368.

Q4: The authors should explain why they only chose Gallic acid (5), Oxypaeoniflorin (21), Paeoniflorin (28), 1,2,3,4,6-O-Pentagalloyl glucose (35), Luteolin (53), Apigenin (55), Benzoylpaeoniflorin (54) and Paeonol (50) to quantify their contents in four ethanolic extracts using HPLC-DAD analysis but not quantify all the 64 compounds.

In this study, though there were 68 compounds found by UFLC-Q-TOF-MS analysis, it was difficult to quantify all the 68 compounds by HPLC-DAD analysis due to the trace contents in the extracts, no/weak UV absorption and no available standards commercially. Moreover, phenols, monoterpenoids and flavonoids had multiple bioactivities, and they were predominant constituents in P. suffruticosa flowers. Therefore, we only quantified their contents in four ethanolic extracts using HPLC-DAD analysis. We have added the sentences in lines 485-489.

Q5: The animal experiment did not contain a positive control group, which made the quality of this data reduced.

We did neglect the positive control group in this animal experiment. We will pay attention to this in the future study.

Round 2

Reviewer 3 Report

I thought this manuscript has enough quality to be published in Antioxidants